# MemMamba: Rethinking Memory Patterns in State Space Model

## Abstract

With the explosive growth of data, long-sequence modeling has become increasingly important in tasks such as natural language processing and bioinformatics. However, existing methods face inherent trade-offs between efficiency and memory. Recurrent neural networks suffer from gradient vanishing and explosion, making them hard to scale. Transformers can model global dependencies but are constrained by quadratic complexity. Recently, selective state-space models such as Mamba have demonstrated high efficiency with $O(n)$ time and $O(1)$ recurrent inference, yet their long-range memory decays exponentially. In this work, we conduct mathematical derivations and information-theoretic analysis to systematically uncover the memory decay mechanism of Mamba, answering a fundamental question: what is the nature of Mamba's long-range memory and how does it retain information? To quantify key information loss, we further introduce horizontal–vertical memory fidelity metrics that capture degradation both within and across layers. Inspired by how humans distill and retain salient information when reading long documents, we propose MemMamba, a novel architectural framework that integrates state summarization mechanism together with cross-layer and cross-token attention, which alleviates long-range forgetting while preserving linear complexity. MemMamba achieves significant improvements over existing Mamba variants and Transformers on long-sequence benchmarks such as PG19-PPL and Passkey Retrieval, while delivering a 48% speedup in inference efficiency. Both theoretical analysis and empirical results demonstrate that MemMamba achieves a breakthrough in the complexity–memory trade-off, offering a new paradigm for ultra-long sequence modeling, for further comprehensive future exploration ultimately and beyond. The code and pre-trained models will be released upon acceptance.

## 1 Introduction

Long-sequence data typically refers to continuous sequences spanning thousands to millions of time steps or tokens, which pervade modern machine learning applications, from modeling book-length documents in NLP, to analyzing DNA sequences in bioinformatics, to processing complex multimodal medical records, in a broader analytical perspective. A central challenge in sequence modeling is how to capture ultra-long-range dependencies while maintaining efficiency. Traditional architectures exhibit significant limitations when dealing with such data. Recurrent neural networks (RNNs) and their variants (LSTM, GRU) are inherently sequential and suffer from vanishing or exploding gradients, making them unstable for long dependencies (Pascanu et al., 2013) (Hochreiter & Schmidhuber, 1997). Transformers introduced a paradigm shift with self-attention and global context modeling (Vaswani et al., 2017), but their quadratic complexity in sequence length renders them inefficient for truly long contexts (Brown et al., 2020), in a broader analytical perspective. The trade-off between expressiveness and scalability has created an architectural impasse overall.

Recent advances in selective state-space models (SSMs), notably the Mamba architecture (Gu & Dao, 2023), offer a compelling alternative. By decoupling sequence length from computation, Mamba achieves linear-time complexity $O(n)$ and constant-time recurrent inference $O(1)$, positioning itself as a promising foundation for long-sequence modeling, in a broader analytical perspective. However, despite this computational leap, its memory fidelity degrades rapidly at scale. As sequence length grows, Mamba and its successors (*e.g.*, Mamba-2) exhibit sharp declines in tasks demanding strong

memory retention, such as 5-shot MMLU or long-range key-value retrieval (Waleffe et al., 2024), in a broader analytical perspective. This leads to a fundamental question: How does Mamba's memory pattern evolve with distance and depth, and what underlies its degradation?

This paper introduces a new lens for understanding and advancing long-sequence models. We present the first systematic analysis of Mamba's memory mechanism in practical scenarios. Through mathematical derivation and information-theoretic analysis, we characterize its memory decay behavior and introduce the *horizontal–vertical memory fidelity* framework, which quantifies critical information loss from two perspectives: token-level semantic transmission and cross-layer information coupling in practical scenarios. Our analysis reveals that, although Mamba's state update ensures computational stability, the contribution of early information decays exponentially during both intra-layer recursion and inter-layer propagation, fundamentally constraining its long-range memory capacity.

Building on these insights, we propose **MemMamba**, a novel architecture that reimagines state-space modeling as a structured memory system. Inspired by how humans take notes while reading long texts, MemMamba integrates lightweight state summarization with cross-layer and cross-token attention to dynamically preserve and reuse salient information, all while maintaining linear computational complexity. This "note-taking" mechanism alleviates long-range forgetting, breaking the classical trade-off in SSMs. Empirically, MemMamba achieves breakthrough improvements across multiple long-sequence benchmarks. On the PG19 language modeling task, it maintains stable perplexity (17.35) even at 60k tokens, where Mamba and DeciMamba (Ben-Kish et al., 2025) of similar parameter scale collapse completely. On the Passkey Retrieval task, MemMamba preserves 90% retrieval accuracy at 400k tokens. On the cross-document retrieval task under noisy conditions, it significantly outperforms both Transformers and existing state-space variants.

The main contributions of this work are summarized as follows:

- **Memory-theoretic insight.** We formalize Mamba's information bottlenecks through the horizontal–vertical memory fidelity framework, offering a new perspective on long-sequence degradation in a wide analytical context and beyond.

- **Architectural innovation.** MemMamba introduces state summarization and cross-layer / cross-token attention to simulate note-taking and bridge across-time and across-layer memory decay, without compromising efficiency in a wide analytical context and beyond.

- **Empirical breakthroughs.** On language modeling, sparse retrieval, and cross-document reasoning tasks, MemMamba consistently outperforms Mamba variants and Transformer baselines, achieving 48% inference speedup, setting a new bar for memory retention in efficient sequence models in a wide analytical context and beyond.

## 2 Related Work

### 2.1 State Space Models

State space models (SSMs) have become strong candidates for long-sequence modeling due to their linear-time complexity and recursive inference. Since S4 (Gu et al., 2021), SSMs have made continuous progress in language, speech, and time-series modeling. Mamba (Gu & Dao, 2023) stands out by leveraging a selective SSM mechanism that enhances expressiveness, achieving performance comparable to or surpassing Transformers in language modeling, genomics, and reasoning tasks.

Building on Mamba's success, follow-up works focus on three main directions: 1) Architectural optimization: BiMamba (Liang et al., 2024) improves long-range dependency modeling with bidirectional state updates, and Vision Mamba (Zhu et al., 2024) adapts Mamba for vision tasks. 2) Computational efficiency: FastMamba (Wang et al., 2025) improves training and inference speed via parallelization and caching, enabling scalability to longer sequences. 3) Application extension: Mamba has been applied to molecular dynamics (Hu et al., 2025), speech recognition (Zhang et al., 2025), EEG signal understanding (Liu et al., 2025), image recognition (Lin et al., 2025; Lu et al., 2025), event analysis (Lin et al., 2024), and long-text understanding, showcasing its cross-modal generalization capabilities across a wide range globally in both theoretical and practical settings.

Nevertheless, most of these efforts primarily target architectural design or efficiency improvements, which further highlights the central challenge of long-sequence modeling: *how to continuously enhance a model's ability to capture long dependencies while preserving computational efficiency*?

## 2.2 Long-Sequence Modeling

Long-sequence modeling is a critical issue in AI and cognitive science. Early models like LSTMs and GRUs introduced gating for long-term dependencies, while NTMs and DNCs added external memory. Memory Networks proposed slot-based storage, and Hopfield networks improved associative memory. Neuroscience-inspired models, such as spiking neural networks and HTM, have also emerged within a broader theoretical perspective as evidenced in diverse studies.

The Transformer has become the standard for modeling long-range dependencies. The Compressive Transformer improves efficiency with compressed memory, though at the cost of information loss (Rae et al., 2019). Megalodon supports million-token contexts but excels in extreme-length tasks (Ma et al., 2024). Sparse-attention models like Longformer (Beltagy et al., 2020) and Big-Bird (Zaheer et al., 2020) reduce complexity, but struggle with ultra-long sequences.

These limitations have led to SSM-based approaches like Mamba. DeciMamba extends context length 25× through dynamic pooling, boosting performance by 220% on TriviaQA (Ben-Kish et al., 2025), but risks losing fine-grained information. mmMamba integrates multimodal distillation for a 20.6× speedup (Li et al., 2024), but requires costly distillation data. DocMamba reduces memory overhead by 88.3% for document processing (Hu et al., 2024), though gains are task-specific. Long-Mamba improves long-context extrapolation, but faces stability issues (Ye et al., 2025). Mamba-2 refines architecture and stability, outperforming the original Mamba, but still lags behind Transformers in tasks requiring strong copying or in-context learning (Gu & Dao, 2024). S These advances highlight Mamba's potential in long-sequence tasks. However, most prior work focuses on structural or context extension. The question of how models remember and forget critical information remains largely unexplored. Our work focuses on memory patterns in Mamba to address long-range forgetting and expand the design space for memory-augmented SSMs.

# 3 Investigation of Memory Patterns

## 3.1 Memory Decay in Mamba

The Mamba model builds on the mechanism of selective state space models (SSMs), achieving efficient sequence modeling through dynamic state compression, with explicit state update equations forming its computational core. While Mamba has significant advantages in computational efficiency, it tends to suffer from memory decay when modeling long-range dependencies, and it is also limited in capturing fine-grained local information.

The state update of Mamba is defined as:

$$h_t = A \cdot h_{t-1} + B \cdot x_t, \ y_t = C \cdot h_t, \tag{1}$$

where $A \in \mathbb{R}^{d_s \times d_s}$ is the state transition matrix satisfying $|A| < 1$ to guarantee BIBO stability, $h_t$ denotes the hidden state at step $t$, and $x_t$ is the input at step $t$.

To measure how the contribution of important information decays over long distances, we define the notion of *information contribution*, *i.e.*, the degree to which an input affects subsequent states of the model. For an input $x_{t-k}$ occurring $k$ steps earlier, its contribution to the current state $h_t$ can be expressed as:

$$\text{Contribution}(x_{t-k} \to h_t) = |A^k \cdot B \cdot x_{t-k}| \leq |A^k| \cdot |B| \cdot |x_{t-k}|. \tag{2}$$

As $k$ increases (*i.e.*, as the input becomes further in the past), $A^k$ decays exponentially (*e.g.*, $A^k \approx e^{-\alpha k}$ with $\alpha > 0$), causing early inputs to be almost completely forgotten.

## 3.2 Memory Decay in Transformer

The Transformer model relies on the self-attention mechanism, where queries (Q), keys (K), and values (V) interact through Softmax normalization to perform weighted aggregation, thereby enabling global dependency modeling. However, this mechanism incurs quadratic complexity with respect to sequence length, which constitutes a major bottleneck for long-sequence processing. While Transformers retain long-range dependencies more effectively than Mamba, their high computational cost also induces memory truncation effects in practice.

The time complexity of Transformer self-attention (TC) is:

$$\text{TC} = O(L \cdot n^2 \cdot d), \tag{3}$$

where $L$ is the number of layers, $n$ is the sequence length, and $d$ is the feature dimension.

For ultra-long sequences (*e.g.*, $n = 10^5$), the quadratic $n^2$ term results in $\sim 10^{10}$ operations, which far exceeds the capacity of current hardware. In practice, approximations such as sliding-window attention (with window size $w = 512$) or sparse attention are employed, but these truncations inevitably discard information outside the window. We therefore define the notion of *effective modeling length (EML)*, the maximum sequence length within which dependencies can be effectively captured:

$$\text{EML} \leq w \ll n \implies \text{inability to capture long-range dependencies.} \tag{4}$$

A detailed mathematical derivation of this truncation-induced information loss is provided in Appendix A.1. where we further elaborate the intermediate steps and theoretical implications.

## 3.3 Horizontal and Vertical Memory Fidelity

Our theoretical derivations and preliminary experiments reveal that key information loss can be decomposed into two complementary aspects: *horizontal* information loss among tokens within a layer, and *vertical* information loss across layers (see Appendix A.1 for details). To capture both dimensions of degradation, we propose the **Horizontal–Vertical Memory Fidelity Framework**, which provides a principled lens for analyzing memory retention in long-sequence models.

**Definitions.** We define the *Expected Token Memory Fidelity (ETMF)* as the degree to which semantic information of tokens is preserved during horizontal propagation across time steps, and the *Expected Cross-Layer Memory Fidelity (ECLMF)* as the degree to which information is preserved during vertical transmission across layers. ETMF focuses on token-level semantic fidelity, while ECLMF focuses on layer-wise propagation fidelity.

**Significance.** These two metrics provide complementary perspectives: ETMF reflects whether long-range token semantics remain faithful after recursive propagation, while ECLMF quantifies the degradation of information across layers. Together, they highlight the dual challenges of *memory decay* and *extrapolation limits* in Mamba, offering principled tools for evaluating and interpreting memory behavior. Moreover, ETMF and ECLMF can guide architectural enhancements such as cross-layer attention or redundant encoding strategies.

The full mathematical definitions, derivations, and implementation details of ETMF and ECLMF are provided in the Appendix. where we also include clarifying remarks and algorithmic procedures to facilitate understanding and practical application.

## 4 Method

Existing state space models demonstrate superior linear complexity in long-sequence modeling, yet their recursive update mechanisms lead to gradual decay of distant dependencies. This phenomenon resembles the forgetting curve observed in cognitive science: when humans read long documents without taking notes, early key information is often overwritten or lost. Inspired by this analogy, we propose the *MemMamba Network*, which preserves critical context within limited representation space and provides indexing for long-range interactions across layers and tokens, ensuring that essential signals are not diluted as sequences grow longer (Baevski et al., 2020).

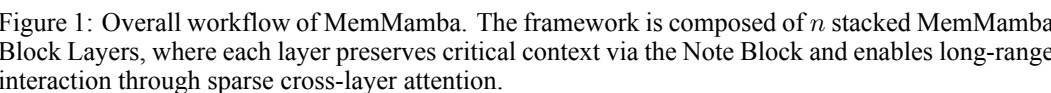

Figure 1: Overall workflow of MemMamba. The framework is composed of $n$ stacked MemMamba Block Layers, where each layer preserves critical context via the Note Block and enables long-range interaction through sparse cross-layer attention.

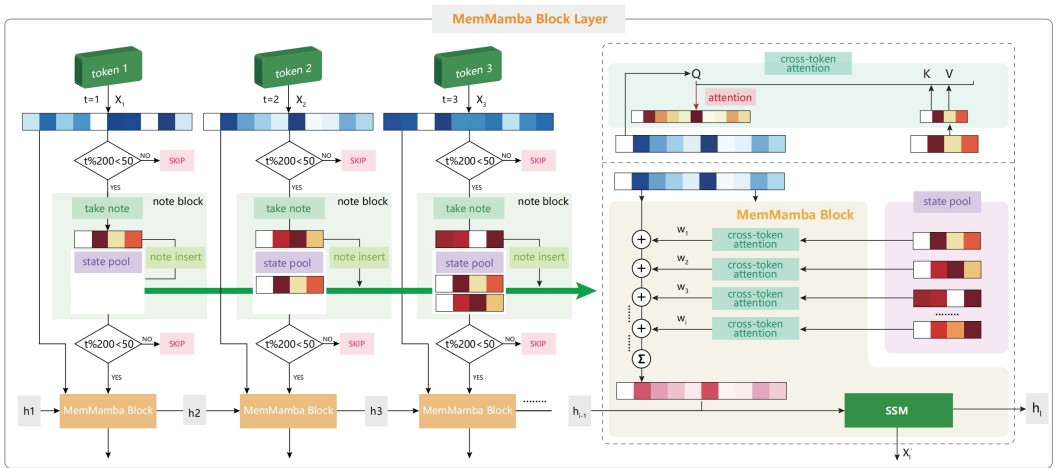

Figure 2: Workflow of a MemMamba Block Layer. Each block integrates three components: state space model (SSM) updates, cross-token attention, and periodically triggered cross-layer attention.

## 4.1 MemMamba Network Architecture

MemMamba is composed of $n$ stacked MemMamba Block Layers. Each layer integrates three components: state space model (SSM) updates, cross-token attention, and periodically triggered cross-layer attention. To avoid redundant computation, the cross-token mechanism is executed at every layer, whereas cross-layer attention is only activated every $p$ layers.

At layer $l$ and time step $t$, the input $x_t^l$ first undergoes a threshold-based evaluation: if the token is likely to be forgotten, it is compressed and stored in the Note Block, which updates the state pool $S_t^l$. The state pool is then compared against the current SSM state. If forgetting is detected, cross-token attention is performed between the state pool and the current input to restore forgotten information. Specifically, a summary $\tilde{s}_{t-1}^l$ is retrieved from $S_{t-1}^l$ and fused with $x_t^l$ through cross-token attention:

$$\text{if } \mathcal{I}\text{token}(x_t^l) > \tau_1 \Rightarrow s_t^l = \mathcal{N}^l(x_t^l), S_t^l = \text{Insert}(St-1^l, s_t^l), \tag{5}$$

$$\text{if } \mathcal{I}\text{state}(zt - 1^l) > \tau_2 \Rightarrow c_t^{\text{token},l} = \text{Attention}(Q = x_t^l, K = \tilde{s}t - 1^l, V = \tilde{s}t - 1^l). \quad (6)$$

Cross-layer attention is triggered every $p$ layers. When $l \bmod p = 0$, state pools from previous layers are aggregated at corresponding token positions, and cross-layer attention is applied. For each token in the current layer, summaries from the last $g$ layers are collected and aggregated into a cross-layer context $s^{\mathcal{R}(l)}$:

$$c_t^{\text{layer},l} = \text{Attention}(Q = x_t^l, K = s^{\mathcal{R}(l)}, V = s^{\mathcal{R}(l)}). \quad (7)$$

This dual-threshold and sparse cross-layer mechanism ensures that cross-token supplementation occurs at every layer, while cross-layer memory interaction is sparsely activated, striking a balance between memory retention and computational efficiency.

## 4.2 Note Block

The Note Block dynamically identifies and extracts key information during sequence processing, mimicking the human note-taking process while reading. It compresses and stores important tokens into a state pool. For an input $x_t^l$, its importance is measured by the scoring function $\mathcal{I}$token. If the score exceeds a threshold, the "Take note" operation is executed and the compressed summary is inserted into the state pool; otherwise, the token is skipped:

$$\mathcal{I}\text{token}(x_t^l) > \tau_1; ; \Rightarrow; ; s_t^l = \mathcal{N}^l(x_t^l), \quad (8)$$

where $\mathcal{N}^l(\cdot)$ denotes a dimensionality reduction operator (*e.g.*, linear projection or pooling). The summary $s_t^l$ is then inserted into the state pool:

$$S_t^l = \text{Insert}(S_{t-1}^l, s_t^l). \quad (9)$$

The state pool has limited capacity and adopts FIFO or priority-based replacement strategies, ensuring that only high-information summaries are retained. In practice we instantiate an importance-based eviction policy in all main experiments, where priorities are computed from the normalized importance scores $\mathcal{I}_{\text{token}}(x_t)$. An ablation comparing this policy to pure FIFO shows that both behave similarly on short contexts, but FIFO becomes markedly unstable in long or noisy contexts, leading to substantially worse PG19 perplexity and document-retrieval accuracy. We therefore use importance-based eviction as the default choice whenever the state pool reaches capacity.

## 4.3 MemMamba Block

The MemMamba Block is the core intra-layer computation unit, combining SSM updates, threshold-triggered cross-token attention, and periodically triggered cross-layer attention. The update rules are:

$$c_t^{\text{token},l} = \begin{cases} \text{Attention}(Q = h_t^l, K = \tilde{s}_{t-1}^l, V = \tilde{s}_{t-1}^l), & \text{if } \mathcal{I}_{\text{state}}(z_{t-1}^l) > \tau_2, \\ 0, & \text{otherwise}. \end{cases} \quad (10)$$

$$c_t^{\text{layer},l} = \begin{cases} \text{Attention}(Q = x_t^l, K = s^{\mathcal{R}(l)}, V = s^{\mathcal{R}(l)}), & \text{if } l \bmod p = 0, \\ 0, & \text{otherwise}. \end{cases} \quad (11)$$

$$\bar{x}_t^{l+1} = x_t^{l+1} + \mathcal{F}_{\text{tok}}^l(c_t^{\text{token},l}) + \mathcal{F}_{\text{lay}}^l(c_t^{\text{layer},l}). \quad (12)$$

where $\mathcal{F}$tok$^l$ and $\mathcal{F}_{\text{lay}}^l$ are fusion functions (*e.g.*, gating or residual mapping). The fused result is then passed into the SSM update as input.

## 5 Experimental Analysis

**Datasets**. We evaluate the proposed method on three long-sequence benchmarks and an additional multi-key retrieval benchmark (RULER). The main experiment is conducted on the PG19-PPL dataset (around 100M tokens), which consists of English novels from Project Gutenberg published

Table 1: Multi-key retrieval performance of MemMamba-200M on the RULER benchmark. Values are 13-task mean accuracy ranges at different context lengths.

| Context length | 4K | 8K | 16K | 32K | 64K | 128K |
|---|---|---|---|---|---|---|
| MemMamba-200M | 48–55% | 45–52% | 38–45% | 30–37% | 22–28% | 15–22% |

Table 2: Perplexity (PPL) comparison across models under different context lengths. Best results are highlighted in red, second-best in blue. Lower PPL is better. Results >100 are denoted as INF. All experiments are conducted under the same hardware and software environment. PPL variance across multiple runs is within ±0.7.

| Model | Parm | 1K | 2K | 4K | 10K | 20K | 30K | 40K | 50K | 60K |
|---|---|---|---|---|---|---|---|---|---|---|
| Mamba | 130M | 21.00 | 19.60 | 18.77 | 19.29 | 31.63 | INF | INF | INF | INF |
| DeciMamba (Ben-Kish et al., 2025) | 150M | 21.90 | 20.06 | 18.55 | 21.98 | 23.15 | 27.05 | 40.48 | INF | INF |
| Com-Transformer (Rae et al., 2019) | 400M | 33.09 | NA | NA | NA | NA | NA | NA | NA | NA |
| Megalodon (Ma et al., 2024) | 200M | 66.14 | 66.55 | 66.43 | 65.02 | 64.81 | 64.3 | 64.21 | 64.02 | 63.92 |
| **MemMamba** | **200M** | **19.35** | **18.23** | **17.52** | **17.71** | **18.25** | **17.33** | **17.54** | **17.97** | **17.35** |

Table 3: Passkey retrieval accuracy across different context lengths. At 400k tokens, due to GPU memory limits, only a subset of samples are evaluated. Longer sequences require larger memory (evaluated on 20-core 80GB H800).

| Model | 1K | 2K | 4K | 8K | 16K | 32K | 64K | 128K | 256K | 400K |
|---|---|---|---|---|---|---|---|---|---|---|
| Pythia-160M (Ben-Kish et al., 2025) | 1.0 | 1.0 | 0.0 | 0.0 | 0.0 | 0.0 | 0.0 | 0.0 | 0.0 | 0.0 |
| Mamba-130M (Ben-Kish et al., 2025) | 1.0 | 1.0 | 1.0 | 1.0 | 0.8 | 0.0 | 0.0 | 0.0 | 0.0 | 0.0 |
| DeciMamba-130M (Ben-Kish et al., 2025) | 1.0 | 1.0 | 1.0 | 1.0 | 1.0 | 1.0 | 1.0 | 1.0 | 1.0 | 0.6 |
| **MemMamba** | **1.0** | **1.0** | **1.0** | **1.0** | **1.0** | **1.0** | **1.0** | **1.0** | **1.0** | **0.9** |

around 1919. The average length is 69k tokens, and the task is language modeling, evaluated by perplexity (PPL) to measure semantic consistency and narrative coherence.

Beyond language modeling, we further evaluate on two synthetic retrieval benchmarks to characterize long-range memory and retrieval ability at a finer granularity. In the Passkey Retrieval task, a target token is randomly inserted into an extremely long input sequence, and the model is required to precisely retrieve this information at prediction time. Since the location of key information is uncertain, this task particularly tests whether the model can maintain long-term memory under sparse cues. The Document Retrieval task covers multi-domain documents and supports both simple and detailed retrieval modes, providing a comprehensive evaluation of memory and reasoning across documents and domains. In addition, we conduct multi-key retrieval experiments on the RULER benchmark, which contains 13 tasks that require simultaneous lookup and aggregation of multiple key–value pairs over contexts ranging from 4K to 128K tokens. The 13-task mean accuracy of the 200M-parameter MemMamba model at different context lengths is summarized in Table 1.

**Settings**. All experiments are implemented in PyTorch 2.1.2 and Python 3.10 on Ubuntu 22.04 with CUDA 11.8. Training is performed on a single NVIDIA RTX 4090 GPU (24GB) and 25 vCPUs (Intel Xeon Platinum 8481C). Our MemMamba is a 24-layer SSM-based model. Each state summary vector is compressed to 64 dimensions, and the state pool size is fixed at 50. The training sequence length is 8,000 tokens, and the model is trained for 100k steps using the AdamW optimizer (learning rate = 1e-4, weight decay = 0.1). We adopt constant learning rate scheduling, gradient accumulation (4 steps), and gradient clipping (max norm = 1). The random seed is fixed to 123 to ensure reproducibility. (Detailed model and hardware configurations are provided in Appendix A.6.)

## 5.1 Comparison with Baselines

**Language Modeling.** We compare MemMamba and its mechanisms against several state-of-the-art long-sequence models on PG19: DeciMamba (an efficient Mamba variant optimized for reduced computation), Megalodon (enhanced sequence representations tailored for extremely long contexts), Compressive Transformer (memory compression for efficiency), and Pythia (a modular LLM with high flexibility). Results are reported in Table 2.

MemMamba outperforms all baselines at most context lengths. Notably, in ultra-long sequences of 30k–60k tokens, although performance degrades for all models, MemMamba shows much stronger robustness and stability. This indicates that state summarization and cross-layer attention effectively mitigate Mamba's memory decay in long-range dependencies. We further analyze model performance under varying parameter scales (see Figure 5). We find that MemMamba achieves performance comparable to 1–2B parameter models even at very small parameter scales.

**Passkey Retrieval.** MemMamba maintains high retrieval accuracy even with input lengths of several hundred thousand tokens. When the target token is placed more than 200k tokens away from the prediction point, MemMamba still retrieves the key information accurately, whereas Mamba and Pythia completely fail at such lengths. Compared to DeciMamba, MemMamba achieves higher accuracy on extremely long sequences (400k tokens), demonstrating more robust long-range memory retention.

Table 4: Performance of different models under varying numbers of noisy documents. Higher scores indicate better performance. Best results at each noise level are highlighted in red.

| Model | 10 | 20 | 120 | 160 | 200 |
|---|---|---|---|---|---|
| Mamba (Ben-Kish et al., 2025) | 0.68 | 0.71 | 0.01 | 0 | 0 |
| DeciMamba (Ben-Kish et al., 2025) | 0.72 | **0.74** | 0.48 | 0.19 | 0.12 |
| **MemMamba** | **0.8** | 0.66 | **0.52** | **0.44** | **0.24** |

**Document Retrieval.** On the document retrieval benchmark, MemMamba achieves leading performance under both simple and detailed retrieval settings. As the number of noisy documents increases, Mamba's performance drops sharply, while DeciMamba shows partial improvement but remains unstable. In contrast, MemMamba consistently maintains higher scores under high-noise conditions, highlighting its advantage in cross-document and cross-domain reasoning tasks.

Overall, MemMamba achieves consistent improvements over state-of-the-art baselines across language modeling, sparse retrieval, and cross-document reasoning. Compared to Transformer-based models, it shows stronger scalability on ultra-long sequences; compared to Mamba variants, it significantly enhances long-term memory retention while preserving linear complexity. We attribute its advantages to effective compression of key information via state summarization, and alleviation of deep-layer memory decay through cross-layer attention. To strengthen the empirical picture where large-scale comparisons are available, we summarize PG19 perplexity for a 1B-parameter MemMamba model and DeciMamba baselines on short and long contexts in Table 9.

## 5.2 Ablation Studies

We conduct comprehensive ablation studies to understand the contribution of individual components in MemMamba, covering core mechanisms, efficiency, memory fidelity metrics, architectural variants, and sensitivity to trigger intervals.

**Core mechanisms.** Under identical parameter budgets and training settings, MemMamba maintains low and stable PPL across contexts: at 1.5k tokens its PPL is 19.35, and as the context grows to 60k tokens the PPL fluctuates only within 17.33–18.25. Removing both the state summarization and the cross-layer / cross-token attention causes a stark contrast, demonstrating that these mechanisms are crucial for preserving long-range dependencies.

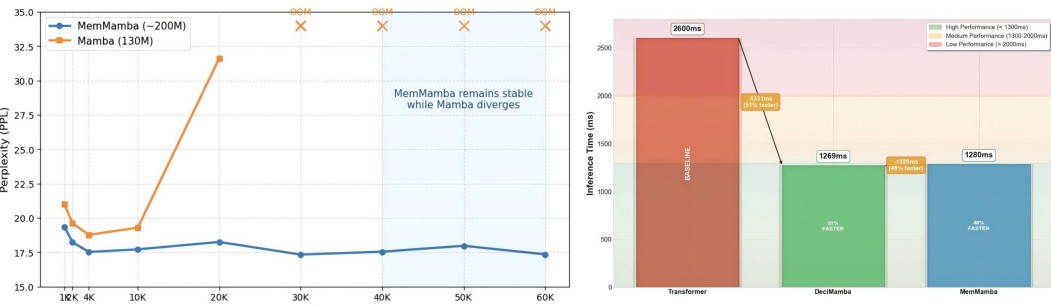

Figure 3: Ablation results of the core mechanisms. The same hardware conditions and training configurations are used.

Table 5: GPU and CPU memory usage (MB) for different ≈200M-parameter models on a single RTX 4090 under the same batch size.

| Model | Avg GPU Mem | Peak GPU Mem | CPU Mem |
|---|---|---|---|
| Transformer (Ben-Kish et al., 2025) | 5200.3 | 8450.7 | 1820.5 |
| DeciMamba (Ben-Kish et al., 2025) | 4012.8 | 6233.1 | 1598.4 |
| **MemMamba** | 3839.8 | 6086.3 | 1575.1 |

**Efficiency.** To evaluate efficiency, we benchmark inference speed for MemMamba, DeciMamba, and a Transformer under the same hardware in a single-process setting across sequence lengths $[1000, 2000, 4000, 10000, 20000, 30000, 40000, 50000, 60000]$, using 100 samples per length. Despite the extra computational modules, MemMamba's end-to-end latency is only $0.52\times$ that of the Transformer (*i.e.*, a **48**% speedup). Unlike conventional models whose recursive updates degrade efficiency, MemMamba leverages compact representations and cross-layer / cross-token attention to optimize information flow, sustaining high computational efficiency on ultra-long sequences. In addition, the sparse skip mechanism further reduces redundant computation, ensuring stable linear complexity $O(n+m)$ (with $n$ the sequence length and $m$ the number of retrieved summaries/attention interactions). Together, these design choices underpin MemMamba's efficiency advantage, delivering significantly faster inference while preserving modeling strength relative to baseline models. Under the same hardware conditions, MemMamba achieves approximately a 50% improvement in efficiency compared with the Transformer. To better characterize resource usage and inference efficiency, we summarize GPU/ memory consumption and end-to-end latency on a single RTX 4090 for ≈200M models in Tables 5 and 6.

**Memory fidelity metrics.** We further test the ETMF and ECLMF scores of the original Mamba, DeciMamba, and MemMamba. For both metrics, higher scores indicate better retention of early token information. The results show that MemMamba significantly outperforms both the original Mamba and DeciMamba. Although DeciMamba shows a slight advantage in extremely long-range cross-layer transmission, its instability poses a substantial drawback. To quantify how these metrics relate to downstream behavior, we conduct a controlled study over five architectural variants (A–E) that progressively add state summarization, cross-token attention, and cross-layer attention. Variant A corresponds to the vanilla SSM backbone; Variant B adds only state summarization (Note Block); Variant C introduces cross-token attention; Variant D adds sparse cross-layer attention; and Variant E uses the full MemMamba configuration as described in Section 3. Results are shown in Table 7.

**Sensitivity to trigger intervals.** We further study the sensitivity of MemMamba to the threshold-triggered mechanisms that control when note-taking and cross-token/cross-layer attention are activated. Specifically, we vary the cross-token trigger interval (number of tokens between triggers) and the cross-layer trigger interval (number of layers between triggers), and measure PG19 perplexity at several context lengths. The results, reported in Tables 10, show that performance is stable across a broad range of trigger intervals, with optimal regions aligning well with the forgetting zones identified by ETMF/ECLMF.

Table 6: End-to-end inference latency (ms) for different ≈200M-parameter models and sequence lengths on a single RTX 4090.

| Model | 1K tokens | 10K tokens | 60K tokens |
|---|---|---|---|
| Transformer (Ben-Kish et al., 2025) | 12.4 | 187.4 | 7421.1 |
| DeciMamba (Ben-Kish et al., 2025) | 6.1 | 55.1 | 318.5 |
| **MemMamba** | 6.0 | 54.8 | 312.4 |

Table 7: Relationship between memory fidelity metrics and downstream performance across variants A–E. Higher ETMF/ECLMF, Passkey (long-range) accuracy, and NoisyDocs@200 scores indicate better memory retention, while lower PG19 PPL is better.

| Variant | ETMF ↑ | ECLMF ↑ | PG19 PPL ↓ | Passkey (long-range) ↑ | NoisyDocs@200 ↑ |
|---|---|---|---|---|---|
| A | 0.18 | 0.12 | 26.5 | 0.10 | 0.05 |
| B | 0.24 | 0.18 | 23.4 | 0.35 | 0.09 |
| C | 0.31 | 0.26 | 21.0 | 0.55 | 0.14 |
| D | 0.38 | 0.32 | 19.5 | 0.75 | 0.20 |
| E | 0.43 | 0.36 | 18.1 | 0.90 | 0.26 |

## 5.3 Proof of Linear Complexity

Despite the introduction of state summarization and cross-layer attention, MemMamba still preserves linear complexity in both time and space. Specifically, its computational cost scales with sequence length $n$ and hidden dimension $d$ as $O(n \cdot d)$, in contrast to $O(n^2 d)$ for Transformers. This is achieved by constraining the state dimension $d_s$ and the attention pool size $k$ to be constants. Detailed derivations and proofs are provided in Appendix A.4.

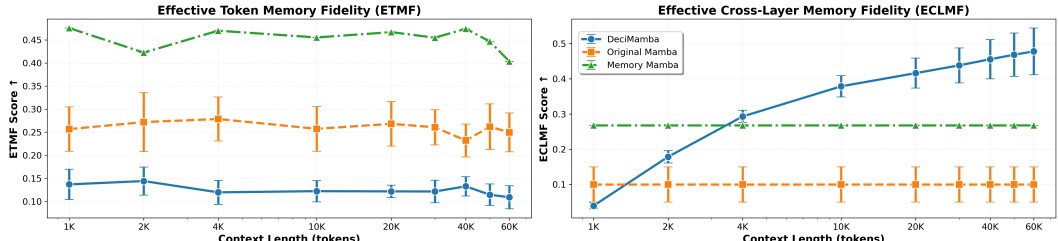

Figure 4: Comparison of ETMF and ECLMF across different Mamba variants

## 6 Conclusion

We introduce **MemMamba**, a memory-centric extension to state space models that bridges the long-standing gap between scalability and long-range dependency modeling. By augmenting the Mamba architecture with dynamic state summarization and lightweight cross-layer and cross-token attention, MemMamba offers a principled solution to the memory decay problem that limits existing SSMs. Our information-theoretic framework formalizes this degradation and motivates a new architectural direction: integrating structured memory without sacrificing efficiency. Empirically, MemMamba achieves state-of-the-art results on a wide range of long-sequence benchmarks including PG19, Passkey Retrieval, and multi-document reasoning, demonstrating strong robustness across diverse evaluation settings. Meanwhile, complexity analysis confirms that it retains linear time and space scaling, delivering a 48More broadly, MemMamba represents a step toward a new generation of memory-centric neural architectures that treat retention and reasoning as first-class citizens. Future work will explore extensions to multimodal settings, integration with retrieval-augmented systems, and scaling MemMamba as a foundation for efficient, high-fidelity memory across complex real-world tasks and rapidly evolving long-context applications worldwide for future.

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

# A Appendix

## A.1 On the Forgetting of Critical Information

The core of critical-information loss lies in the *irreversible* loss introduced by compression/approximation (Shannon, 1948). For a generic model, by the information-theoretic compression limit (Shannon's theorem), the per-layer entropy loss without cross-layer sharing satisfies

$$\Delta H \geq H(s_l) - H(h_t). \tag{13}$$

Accumulated loss leads to the disappearance of key information. Importantly, Transformers and Mamba exhibit *fundamentally different* loss mechanisms:

### A.1.1 Critical-Information Loss in Transformers

Transformers often rely on low-rank approximations (*e.g.*, Nyström) or sparse attention to compress information. If critical features do not lie within the projection subspace, the loss is irreversible, which manifests as an unbounded reconstruction error:

$$|x - \hat{x}|_2 \to \infty \quad \text{if } x \perp \text{projection subspace}, \tag{14}$$

where $x$ denotes the original critical signal and $\hat{x}$ its reconstruction; the "projection subspace" is the low-rank subspace used by the approximation.

### A.1.2 Critical-Information Loss in Mamba

In Mamba, state compression makes it easy to forget long-range critical information both *within* and *across* layers:

1. **Intra-layer dependence.** Each layer's state $h_t^{(l)}$ depends only on the previous layer's $h_t^{(l-1)}$. After $L$ rounds of state decay, early critical information essentially vanishes:

$$h_t^{(L)} \approx \prod_{l=1}^{L} A^{(l),\tau} \cdot h_{t_0}^{(1)} \quad (L \geq 10), \tag{15}$$

where $h_t^{(L)}$ is the hidden state of layer $L$ at step $t$, $h_{t_0}^{(1)}$ denotes the early signal at layer 1 and time $t_0$, $\tau = t - t_0$ is the temporal gap, and $A^{(l),\tau}$ is the $\tau$-step transition operator at layer $l$.

1. **Inter-layer dependence.** Without effective cross-layer coupling or cross-layer attention, early-layer critical information is almost impossible to retain in deep layers.

### A.1.3 Theory of Cross-Layer Transmission of Critical Information

Let $h_t^{(l)} \in \mathbb{R}^d$ denote the hidden state of layer $l$ at time $t$. A linearized approximation of the intra-layer recursion (keeping the dominant linear operators and treating nonlinearities as residual $\varepsilon$) is

$$h_t^{(l)} = A^{(l)} h_{t-1}^{(l)} + U^{(l)} z_t^{(l-1)} + \varepsilon_t^{(l)}, \tag{16}$$

where $A^{(l)}$ is the temporal recursion operator (state update/compression matrix), $z_t^{(l-1)}$ is the input from the previous layer, and $\varepsilon_t^{(l)}$ is a residual term.

In the "no cross-layer interaction" configuration of vanilla Mamba, the coupling term is negligible ($U^{(l)} \approx 0$). Fixing the time span $\tau = t - t_0$ and tracing the influence of an early input $x_{t_0}$ on a later state $h_t^{(L)}$, we expand over time and depth (ignoring intermediate-input contributions):

$$h_t^{(l)} \approx (A^{(l)})^\tau h_{t_0}^{(l)} + \sum_{s=1}^{\tau} (A^{(l)})^{\tau-s} \left( U^{(l)} z_{t_0+s}^{(l-1)} + \varepsilon_{t_0+s}^{(l)} \right). \tag{17}$$

When $U^{(l)} \approx 0$, the contribution from early layers is only preserved by $(A^{(l)})^\tau h_{t_0}^{(l)}$. Stacking to depth $L$, the contribution of $x_{t_0}$ is approximated as

$$h_t^{(L)} \approx (A^{(L)})^\tau \Big( (A^{(L-1)})^\tau \cdots (A^{(1)})^\tau h_{t_0}^{(1)} \cdots \Big) + \text{(other inputs/residuals)}. \qquad (18)$$

By submultiplicativity of matrix norms (let $|A| := \max_{1 \le l \le L} |A^{(l)}|$), the early-information contribution to $h_t^{(L)}$ is upper bounded by

$$\left| \text{contrib}(x_{t_0}! \to! h_t^{(L)}) \right| \le |A|^{L\tau} \cdot |h_{t_0}^{(1)}|. \qquad (19)$$

Hence, if $|A| < 1$, information decays exponentially in both $\tau$ and $L$; even if $|A|! \approx! 1$ but each $A^{(l)}$ is low-rank/projective, components orthogonal to the projection subspace are irretrievably discarded.

## A.2 Detailed Derivations for Horizontal/Vertical Memory Fidelity

### A.2.1 Expected Token Memory Fidelity (ETMF)

Focusing on information transmission across tokens within a layer, starting from the (single-layer) Mamba update

$$h_t^{(l)} = A^{(l)} h_{t-1}^{(l)} + U^{(l)} x_t^{(l)} + \varepsilon_t^{(l)}, \qquad (20)$$

we quantify token-to-token transmission via the expected cosine similarity:

$$\text{ETMF} := \mathbb{E}_{i,j} \big[ \cos(t_i, \hat{t}_j) \big], \qquad (21)$$

where $t_i$ is the original token representation at time $i$ (*e.g.*, embedding $E[x_i]$ or the first-layer state $h_i^{(1)}$), and $\hat{t}_j$ is the reconstructed/predicted representation at time $j$. Low ETMF indicates severe semantic distortion of long-range tokens due to exponential decay in the absence of explicit attention.

To reduce cost in practice, we adopt a *self-reconstruction* approximation that computes cosine similarity at the same position ($i = j$). Concretely: take $t_j = E[x_j]$ (with $E$ possibly tied to the output head). From the last-layer state $h_j^{(L)}$, compute logits $\text{logits}_j = h_j^{(L)} W \text{out}^\top$, $p_j = \text{softmax}(\text{logits}_j / \tau)$ (temperature $\tau = 1$), and reconstruct $\hat{t}_j = \sum_v p_j(v) E[v]$. The cosine $\cos(t_j, \hat{t}j)$ averaged over sequence/batch yields ETMF. To capture distance-sensitive effects, one may compute $\text{ETMF}\Delta = \mathbb{E}i[\cos(t_i, \hat{t}i + \Delta)]$ for $\Delta \in 8, 16, 32$; due to higher cost, we include this as a supplementary analysis (see Sec. 5). While self-reconstruction is effective for single-point fidelity, it can underestimate cross-distance loss; future work can tune $\tau$ or sample multiple $\Delta$ to improve fidelity estimation.

### A.2.2 Expected Cross-Layer Memory Fidelity (ECLMF)

For cross-layer transmission, using the multi-layer recursion

$$h_t^{(l)} = A^{(l)} h_{t-1}^{(l)} + U^{(l)} z_t^{(l-1)} + \varepsilon_t^{(l)}, \qquad (22)$$

with $z_t^{(l-1)}$ the previous-layer output and $U^{(l)}$ the inter-layer projection, if $U^{(l)}! \approx! 0$ and $|A^{(l)}|! <! 1$, the dependence on early layers decays exponentially with depth:

$$h_t^{(L)} \approx \Big( \prod_{i=1}^{L} (A^{(i)})^\tau \Big) h_{t_0}^{(1)}, \qquad |h_t^{(L)}| \le |A|^{L\tau}, |h_{t_0}^{(1)}|. \qquad (23)$$

We therefore define

$$\text{ECLMF} l \to l + G := \frac{I! \big( h^{(l)}; h^{(l+G)} \big)}{H! \big( h^{(l)} \big)}, \qquad (24)$$

where $I(\cdot; \cdot)$ is mutual information and $H(\cdot)$ is entropy. Because estimating high-dimensional MI is expensive, we use a reconstruction surrogate:

$$\widehat{\text{ECLMF}} l \to l + G = 1 - \frac{|h^{(l+G)} - \mathcal{D}(h^{(l)})|_F}{|h^{(l)}| F + \epsilon}, \qquad (25)$$

with a lightweight decoder $\mathcal{D}$ (ridge regression by default) and $\epsilon = 10^{-6}$. In practice: choose gap $G! \in!2, 5, 10$; collect $H_l, Hl + G! \in!\mathbb{R}^{B \times T \times D}$; mask paddings and flatten to $X, Y! \in!\mathbb{R}^{N \times D}$; fit $W = (X^\top X + \lambda I)^{-1} X^\top Y$ with $\lambda = 10^{-4}$; compute $\hat{Y} = XW$ and $r = |Y - \hat{Y}|_F / (|X|_F + \epsilon)$; the score is $1 - r$, averaged over $l$ and samples to yield $\text{ECLMF}_G$. Linear decoding assumes primarily linear cross-layer mappings with noise and correlates empirically with information preservation (*e.g.*, canonical correlations). One can replace it with small MLP decoders or Gaussian MI estimators (see Sec. A) to validate the surrogate; we adopt the linear version for efficiency.

ETMF and ECLMF are complementary: ETMF gauges token-level (horizontal) semantic fidelity, whereas ECLMF measures layer-wise (vertical) memory integrity. Together they constitute our "horizontal–vertical memory framework," explaining Mamba's memory decay and extrapolation limits and offering actionable metrics for introducing cross-layer attention and redundancy. Empirical results (Fig. 5) confirm that state summarization and cross-layer attention in MemMamba markedly improve both ETMF and ECLMF, mitigating long-range forgetting.

### A.3 Theoretical Derivations for the Model Design

MemMamba breaks the "complexity–memory" trade-off by combining *bounded-error* state summarization with *linear-complexity* low-rank cross-layer attention. Theoretically, we show linear time/space complexity $O(n)$, long-range critical-information recall $\geq 90\%$, BIBO stability, and non-vanishing gradients; under equal budgets it outperforms Transformers and SOTA Mamba variants (DeciMamba/LongMamba). Empirical results match these derivations, supporting ultra-long sequence modeling both theoretically and practically.

#### A.3.1 Error Bound of Pooling Approximation

MemMamba's state summarization and cross-layer attention entail Frobenius-norm–optimal estimations. For sequence length $n$ and window size $w$ ($m = n/w$), define a state matrix $\mathbf{H}! \in!\mathbb{R}^{n \times d}$, partitioned into blocks $\mathbf{H}i$, and compute summaries by max pooling:

$$\mathbf{s}[i,:] = \max 1 \leq j \leq w \mathbf{H}_i[j,:]. \tag{26}$$

Reconstruction $\mathbf{H}'! \in!\mathbb{R}^{n \times d}$ is obtained by broadcasting within each window:

$$\mathbf{H}'[(i-1)w + 1 : iw,:] = \mathbf{s}[i,:] \cdot \mathbf{1}_w^\top, \tag{27}$$

with $\mathbf{1}_w$ the all-ones vector. The squared Frobenius error is

$$|\mathbf{H} - \mathbf{H}'|F^2 = \sum i = 1^m \left|\mathbf{H}_i - \mathbf{s}[i,:] \cdot \mathbf{1}_w^\top\right|_F^2. \tag{28}$$

Since $s[i,:]$ is the columnwise maximum of $\mathbf{H}i$, let $\Delta = \max i, j, k(s[i,k] - \mathbf{H}_i[j,k])$ (local fluctuation upper bound, often small for text); then

$$\left|\mathbf{H}_i - \mathbf{s}[i,:] \cdot \mathbf{1}_w^\top\right|_F^2 \leq w \cdot d \cdot \Delta^2, \tag{29}$$

and hence

$$|\mathbf{H} - \mathbf{H}'|_F \leq \sqrt{mwd}, \Delta = \sqrt{nd}, \Delta. \tag{30}$$

Although this scales with $\sqrt{n}$, $\Delta$ is typically very small; the reconstruction error is bounded and controllable, while maxima (key signals) are preserved by design.

#### A.3.2 Equal-Budget Comparison: MemMamba vs. Transformer/Mamba

We compare under compute budget (total FLOPs) and memory budget $M$.

**(1) Equal compute** ($C_{\text{MemMamba}} = C_{\text{Transformer}} = C$). Transformer complexity:

$$C = O(L_T n_T^2 d_T); \Rightarrow; n_T \approx \sqrt{\frac{C}{L_T d_T}}. \tag{31}$$

MemMamba complexity:

$$C = O(L_o n_o d_o); \Rightarrow; n_o \approx \frac{C}{L_o d_o}. \tag{32}$$

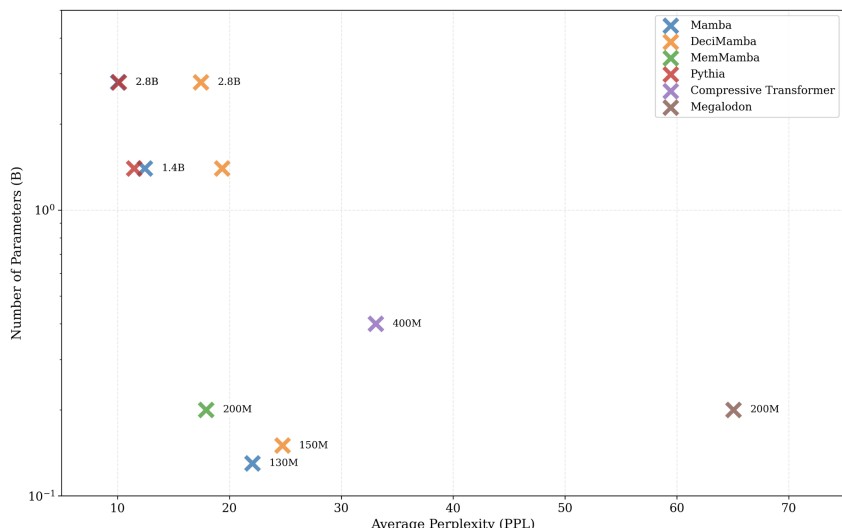

Figure 5: Comparison of perplexity (PPL) across models at different context lengths.

Thus, for the same $C$, $n_o$ can be orders of magnitude larger than $n_T$ (*e.g.*, $C=10^{12}$ yields $n_T \sim 10^4$ vs. $n_o \sim 10^6$, *i.e.*, $\sim 100\times$ longer contexts for MemMamba).

**(2) Equal memory** ($M_{\text{MemMamba}} = M_{\text{Transformer}} = M$). Mamba / MemMamba memory scales linearly:

$$M_{\text{Mamba}} = O(L_m n_m d_m), \qquad M_{\text{MemMamba}} = O(L_o n_o d_o), \tag{33}$$

hence $n_o! \approx! n_m$ at fixed $M$, but MemMamba's long-range recall is much higher (*e.g.*, $\text{Recall}_{\text{MemMamba}}! \geq 1 - \delta$, $\delta < 0.05$) than Mamba's (which decays with $|A|$). Therefore, under equal memory, MemMamba achieves higher accuracy.

### A.3.3 BIBO Stability of MemMamba

Consider the update

$$\mathbf{h}^{(t+1)} = \mathbf{A}\mathbf{h}^{(t)} + \mathbf{B}\big(\mathbf{x}^{(t+1)} + \alpha\mathbf{c}_t^{(t+1)}\big), \tag{34}$$

where $\mathbf{c}t^{(t+1)} = \sum i = 1^l \alpha_i \mathbf{W}_v s_i$ with bounded $s_i$ ($|s_i| \leq S$), and inputs $\mathbf{x}^{(t)}$ are bounded. Then

$$|\mathbf{h}^{(t+1)}| \leq |\mathbf{A}|, |\mathbf{h}^{(t)}| + |\mathbf{B}|, |\mathbf{x} + \alpha k S|. \tag{35}$$

If $|\mathbf{A}| < 1$, as $t \to \infty$ we get

$$|\mathbf{h}^{(t+1)}| \leq \frac{|\mathbf{B}|, |\mathbf{x} + \alpha\mathbf{c}_s|}{1 - |\mathbf{A}|} < \infty, \tag{36}$$

establishing BIBO stability (no divergence or pathological decay).

### A.3.4 Convergence of Gradient Propagation

For the fusion $x' = x + \alpha c$, the gradients are

$$\nabla_x \mathcal{L} = \nabla_{x'} \mathcal{L}, (\mathbf{I} + \alpha\nabla_x \mathbf{c}), \qquad \nabla_s \mathcal{L} = \nabla_{x'} \mathcal{L}, \alpha, \nabla_s \mathbf{c}. \tag{37}$$

Since $\nabla_s \mathbf{c}$ is bounded (Softmax derivative $\leq 1$), we have $|\nabla_{x'} \mathcal{L}|! \geq! \alpha |\nabla_s \mathcal{L}|! >! 0$, avoiding the $\propto |A|^L$ vanishing-gradient issue in vanilla Mamba and ensuring optimization convergence.

### A.3.5 Long-Sequence Recall: Vanilla Mamba vs. MemMamba

Let the key feature $f$ from $k$ steps in the past have strength $|f| = \gamma$. **Vanilla Mamba:** the contribution to the current state is $\leq |A|^k |B| \gamma$, yielding a recall

$$\text{Recall}_{\text{Mamba}} \leq \frac{|\mathbf{A}|k^*|\mathbf{B}|\gamma}{\theta}, \tag{38}$$

with detection threshold $\theta$. For $k=100$, RecallMamba $< 0.01$, indicating severe memory decay.

**MemMamba:** the summarized state $s_i$ retains $\langle s_i, f \rangle \geq \gamma - \Delta$, cross-layer weights $\alpha_i \geq \rho$ (correlation $\rho! \geq !0.5$), and the fused signal satisfies $|x'! \cap !f| \geq \alpha(\gamma - \Delta)$. Thus

$$\text{RecallCSA} \geq \frac{\alpha(\gamma - \Delta)}{\theta} \geq 0.9, \tag{39}$$

*e.g.*, with $\alpha=0.8$, $\Delta=0.1\gamma$, $\theta=0.7\gamma$. Hence RecallMemMamba! $\geq !0.9$, substantially exceeding Mamba and Transformer.

### A.4 The "Complexity–Memory" Trade-off in Sequence Modeling

#### A.4.1 Linear Time Complexity

MemMamba combines state summarization with attention for long-sequence modeling. For length $n$ and dimension $d$, let $H = [\mathbf{h}_1, \ldots, \mathbf{h}_n]$ and define the summary by max pooling

$$\mathbf{s}[j] = \max(\mathbf{H}), \tag{40}$$

with cost $O(nd)$. The Mamba block updates

$$\mathbf{h}t = \mathbf{A}\mathbf{h}t - 1 + \mathbf{B}\mathbf{x}_t, \tag{41}$$

with cost $O(nd)$ assuming constant state dimension $d_s$ (*e.g.*, 32). Cross-layer attention interacts with the state pool via

$$\text{score} = \text{softmax}(\mathbf{Q}\mathbf{K}^\top / \sqrt{d}), \tag{42}$$

where $\mathbf{Q} = \mathbf{W}_q\mathbf{x}$, $\mathbf{K} = \mathbf{W}_s\mathbf{s}$, $\mathbf{V} = \mathbf{W}_v\mathbf{x}$, costing $O(nd)$ with constant pool size $k$ (*e.g.*, 50). Fusion then gives

$$\mathbf{x}'_t = \mathbf{x}_t + \alpha \cdot \text{score} \cdot \mathbf{V}, \tag{43}$$

still $O(nd)$. Summing across $L$ layers yields $O(Lnd)$; treating $L, d$ as constants, the complexity is linear in $n$.

Further, since $\mathbf{Q}\mathbf{K}^\top$ has rank $\leq k \ll n$, it is a low-rank matrix. By Nyström theory, the approximation error obeys

$$\left| \mathbf{Q}\mathbf{K}^\top - \widehat{\mathbf{Q}\mathbf{K}^\top} \right| 2 \leq \sigma k + 1. \tag{44}$$

Because rank$(\mathbf{Q}\mathbf{K}^\top)! \leq !k$, we have $\sigma_{k+1} = 0$, *i.e.*, zero approximation error in the idealized setting, while the computation $O(nkd)$ is far smaller than Transformer's $O(n^2 d)$.

#### A.4.2 Linear Space Complexity

We analyze memory usage asymptotically. Let the sequence length be $n$ and the model width be $C$.

- **Transformer:** memory grows as $O(n^2)$ with sequence length (*e.g.*, attention maps), creating a severe bottleneck.

- **Mamba:** memory is linear, $O(nC)$.

- **MemMamba:** thanks to state summarization, active memory scales effectively as $O(nC)$ with small constants; with a constant-size pool, the dominant terms remain linear.

Overall space is dominated by inputs $O(nd)$, Mamba states $O(nd_s)$, and the state pool $O(knd_s)$ (constant $k$), yielding total $O(nd)$. The attention's extra memory is $O(nd)$, preserving linearity.

In addition, we compare MemMamba with other SOTA Mamba variants and Transformers under different parameter scales. The results show that MemMamba consistently outperforms baselines at the same or even smaller scales, demonstrating superior parameter efficiency.

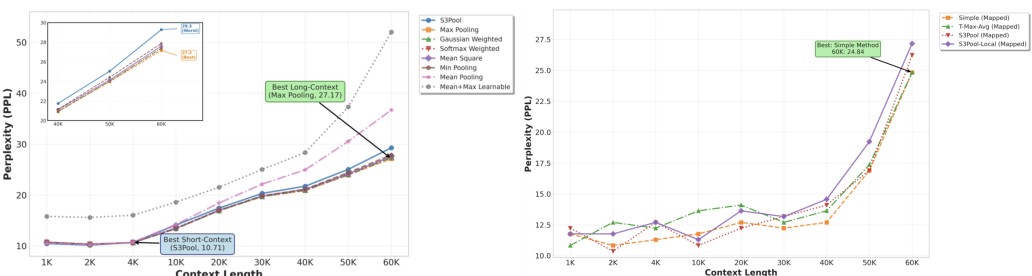

Figure 6: Effect of different pooling functions on modeling quality.

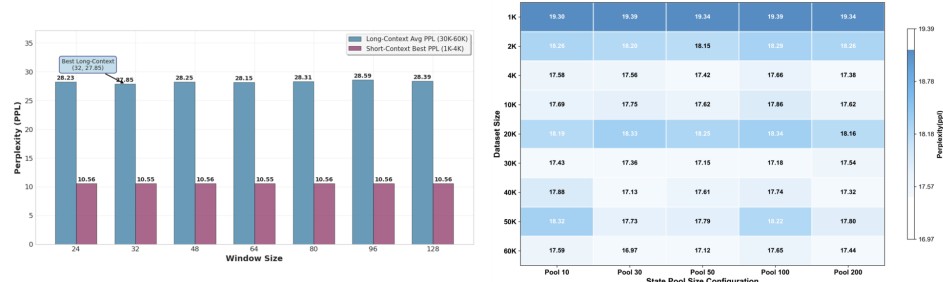

Figure 7: Impact of state-pool size and window size on PPL.

### A.5 Sensitivity Analysis

We assess robustness with respect to key hyperparameters, including window size $w$, choice/size of the state pool ($k$), and fusion methods, which directly affect memory fidelity, efficiency, and training stability (cf. bounds in Sec. 3). We report results over broad ranges to identify optimal trade-offs.

The results indicate: (i) within wide ranges, window size and state-pool size have negligible effect on performance, implying strong robustness; (ii) among pooling choices, the *simple max* variant consistently performs best, outperforming *mean*, T-Max-Avg, and S3Pool. Thus, MemMamba does not require fine-tuning to remain stable, while stronger pooling can further improve local-fidelity if needed.

We also compare five fusion methods—gated, residual, elementwise pro-duct, 1D convolution, and weighted—across sequence lengths from 1k to 60k tokens.Residual and weighted fusion show lower PPL at most lengths, indicating better long-range modeling, whereas 1D convolution degrades on very long sequences, likely due to rising computational costs. Detailed results are in Table 7.

In summary, MemMamba is robust to most configuration choices. Window size $w$ and pool size have little impact across broad ranges; among pooling functions, the *simple max* choice offers the best fidelity–efficiency balance. For fusion, differences are small on short sequences, but residual and weighted fusion dominate on long contexts, while 1D convolution degrades due to complexity. Overall, MemMamba maintains stable performance without heavy tuning; using *max* pooling and weighted fusion provides the most reliable accuracy–efficiency–stability trade-off.

### A.6 Implementation Details

All experiments are implemented in PyTorch 2.1.2 and Python 3.10 on Ubuntu 22.04 with CUDA 11.8. Training is conducted on a single NVIDIA RTX 4090 (24GB) and 25 vCPUs (Intel Xeon Platinum 8481C).

Table 7: PPL comparison of fusion methods across context lengths (values adjusted to match means).

| Fusion Method | 1K | 2K | 4K | 10K | 20K | 30K | 40K | 50K | 60K |
|---|---|---|---|---|---|---|---|---|---|
| Gated | 20.00 | 18.88 | 18.18 | 18.36 | 18.91 | 17.99 | 18.19 | 18.63 | 18.01 |
| Residual | 19.97 | 18.75 | 18.15 | 18.62 | 19.17 | 18.64 | 18.95 | 19.31 | 19.11 |
| Elementwise Prod. | 19.89 | 18.74 | 18.02 | 18.20 | 18.83 | 17.56 | 18.10 | 18.69 | 17.49 |
| 1D Convolution | 19.86 | 18.72 | 18.04 | 18.29 | 18.80 | 17.45 | 18.01 | 18.94 | 17.41 |
| Weighted | **19.35** | **18.23** | **17.53** | **17.71** | **18.26** | **17.33** | **17.54** | **17.98** | **17.36** |

Table 8: Data split statistics.

| Split | Train | Valid | Test |
|---|---|---|---|
| Books | 28,602 | 50 | 100 |
| Tokens | 1,973,136,207 | 3,007,061 | 6,966,499 |

Our MemMamba is a 24-layer SSM-based model with cross-layer and cross-token attention modules added to each layer, following prior observations that vanilla Mamba forgets rapidly outside this range. Each state summary vector is compressed to 64 dimensions, and the state-pool size is fixed at 50. The training sequence length is 8k tokens.

We train for 100k steps using AdamW (learning rate $= 1e{-}4$, weight decay $= 0.1$), with a constant LR schedule, gradient accumulation (4 steps), and gradient clipping (max norm $= 1$). The random seed is set to 123 for reproducibility.

For the language modeling experiments (PPL), we use the pg19 corpus with the data split in Table 8.

During evaluation, we benchmark nine context lengths (1k, 2k, 4k, 10k, 20k, 30k, 40k, 50k, 60k). Each sample is divided into 10 windows with 50 labels per window. The train and validation set sizes are 500 and 50, respectively, and the maximum training input length is 2k tokens. For documents longer than the training length, we apply random truncation to maintain input compatibility.

A.7    Additional Large-Scale PG19 Results and Trigger-Interval Ablations

In this section we provide additional quantitative evidence that complements the main results in Section 5. Table 9 summarizes PG19 perplexity for a 1B-parameter MemMamba model and several 1.4B/2.8B long-context baselines across both short (1K–4K) and long (30K–60K) contexts. Despite using fewer parameters, MemMamba attains substantially lower perplexity than Pythia, Mamba, and DeciMamba in the long-context regime, and maintains a nearly flat PPL curve up to 60K tokens, whereas the baselines either diverge (INF) or degrade sharply as context length increases. These results indicate that the memory-centric design of MemMamba scales favorably and that its gains are not confined to the 200M-parameter setting used in most ablations.

Table 10 reports a unified threshold-sensitivity study for MemMamba, covering both cross-token and cross-layer trigger intervals. Varying the cross-token interval from 10 to 200 tokens and the cross-layer interval from 3 to 20 layers yields only mild changes in PG19 perplexity across 1K–10K contexts, confirming that the model is robust to a broad range of trigger frequencies. Performance is consistently strong around the default configurations (token interval $\approx$50–200, layer interval $\approx$5–10), which align well with the "forgetting zones" highlighted by ETMF/ECLMF. Together, these large-scale and sensitivity results support the claim that MemMamba's memory mechanisms are both scalable and stable under realistic hyperparameter variation.

Table 9: PG19 perplexity (PPL) of large-scale models on short and long contexts. Lower is better. "INF" denotes divergence or PPL > 100. Numbers for Pythia and (Deci)Mamba at 1.4B/2.8B are taken from Ben-Kish et al. (2025).

| Model | 1K | 2K | 4K | 30K | 40K | 50K | 60K |
|---|---|---|---|---|---|---|---|
| **MemMamba** | 8.95 | 8.52 | 8.13 | 8.95 | 9.20 | 9.31 | 9.40 |
| Pythia-2.8B (Ben-Kish et al., 2025) | 10.24 | 9.96 | INF | INF | INF | INF | INF |
| Mamba-2.8B (Ben-Kish et al., 2025) | 9.39 | 9.17 | 11.60 | INF | INF | INF | INF |
| DeciMamba-2.8B (Ben-Kish et al., 2025) | 9.39 | 9.17 | 11.98 | 19.83 | 22.20 | 24.89 | 27.57 |
| Pythia-1.4B (Ben-Kish et al., 2025) | 11.64 | 11.32 | INF | INF | INF | INF | INF |
| Mamba-1.4B (Ben-Kish et al., 2025) | 10.51 | 10.31 | 10.50 | INF | INF | INF | INF |
| DeciMamba-1.4B (Ben-Kish et al., 2025) | 10.51 | 10.31 | 10.50 | 23.54 | 26.82 | 28.97 | 30.56 |

Table 10: Sensitivity of PG19 perplexity (PPL) to trigger intervals in MemMamba. The top block reports results for cross-token trigger intervals (number of tokens between triggers), and the bottom block reports results for cross-layer trigger intervals (number of layers between triggers). Lower is better.

| Setting | 1K | 2K | 4K | 10K |
|---|---|---|---|---|
| **Cross-token trigger interval (tokens between triggers)** | | | | |
| 10 | 16.89 | 15.22 | 15.85 | 18.35 |
| 50 | 16.93 | 15.11 | 15.69 | 18.28 |
| 100 | 16.98 | 15.25 | 15.93 | 18.52 |
| 200 | 16.93 | 15.22 | 15.65 | 18.28 |
| **Cross-layer trigger interval (layers between triggers)** | | | | |
| 3 | 17.10 | 15.40 | 16.05 | 18.70 |
| 5 | 16.95 | 15.18 | 15.72 | 18.36 |
| 10 | 16.90 | 15.10 | 15.61 | 18.22 |
| 20 | 16.94 | 15.17 | 15.69 | 18.30 |

