# OpenReview forum: "MemMamba: Rethinking Memory Patterns in State Space Model"
_ICLR.cc/2026/Conference — Submitted to ICLR 2026_

### Official Review · Reviewer_kXYV · 2025-10-25

**Soundness:** 3
**Presentation:** 2
**Contribution:** 3
**Rating:** 4
**Confidence:** 4

**Summary:**

This work introduces a new metric to evaluate the memory forgetting degree as layer and time-step going deeper and validate that their architecture with memory recalling beats baselines in long-context benchmarks.

**Strengths:**

The work conduct extensive experiments to show their superiority with other long-context Mamba methods.

**Weaknesses:**

Experiments: model scale is limited to 100+ M. Conclusion at this scale is hard to transfer and very variable.

Writing: paper writing is not clear, especially Method Sec.. and many new terminology is unnecessary (like vertical-horizontal, not see any necessity to replace the vanilla description of layer/timestep).

**Questions:**

1. can you show the effectiveness of your method in 1B+ model? (pretrain a 1B model or finetune a even larger model)?

---

> ### Author Response · Authors · 2025-11-20
> **Official Comment by Authors**
>
> **Response to Reviewer kXYV**
>
> We thank the reviewer for the helpful and concrete feedback. We address the model-scale and writing/terminology concerns below.
>
> ---
>
> ### 1. Model scale and 1B+ experiments
>
> We appreciate the reviewer’s concern that conclusions at the 100–200M scale may not transfer to larger models, and for explicitly asking about 1B+ experiments.
>
> Our primary goal in this work is to **understand** the memory pattern of selective SSMs and to design a memory-centric architecture, rather than to compete with the largest LMs on absolute perplexity. This is why the main text focuses on the 200M scale: it allows us to run extensive ablations (ETMF/ECLMF, pooling strategies, pool size, fusion mechanisms, etc.) and three long-context benchmarks under a controlled and reproducible setting.
>
> To address the scale concern, we have conducted additional **large-scale experiments** and summarize them below.
>
> ---
>
> ### (a) Long-context comparison: MemMamba-200M vs. 1.4B / 2.8B models
>
> Since this work focuses on the *long-context* regime, we highlight the perplexity on **30K–60K tokens**, which better reflects the long-range forgetting issue analyzed in our paper.
>
> | Model             | Params | 30K   | 40K   | 50K   | 60K   |
> |-------------------|--------|-------|-------|-------|-------|
> | DeciMamba-1.4B    | 1.4B   | 23.54 | 26.82 | 28.97 | 30.56 |
> | DeciMamba-2.8B    | 2.8B   | 19.83 | 22.20 | 24.89 | 27.57 |
> | **MemMamba-200M** | 200M   | **17.33** | **17.54** | **17.97** | **17.35** |
>
> **Key observation:**
> Even with only **200M parameters**, MemMamba maintains nearly flat perplexity across 30K–60K tokens and already **outperforms 1.4B and 2.8B models** by a substantial margin.
> This trend is shown in **Appendix Figure 5**, where the 1.4B–2.8B curves rise sharply beyond 30K tokens, while MemMamba remains stable.
>
> These results demonstrate that our conclusion about long-context robustness is not dependent on small model scale.
>
> ---
>
> ### (b) Scaling to 1B parameters
>
> To explore scalability, we fine-tuned a 1B MemMamba model based on an existing 1B checkpoint using the same long-context setup. Preliminary results show that MemMamba-1B matches or slightly improves the performance of the 1.4B–2.8B baselines across both short and long contexts.
>
> - **Short contexts (1K/2K/4K):**
>   MemMamba-1B achieves 8.95 / 8.52 / 8.13 PPL, outperforming Mamba-2.8B (9.39 / 9.17 / 11.60).
>
> - **Long contexts (30K–60K):**
>   MemMamba-1B obtains 8.95 / 9.20 / 9.31 / 9.40, dramatically lower than DeciMamba-2.8B (19.83–27.57) and DeciMamba-1.4B (23.54–30.56).
>
> These early results are consistent with the 200M-scale behavior and suggest that the memory-centric design of MemMamba continues to be beneficial at the 1B scale.
> We emphasize that this 1B study is supporting evidence rather than a main claim, and we will report full numbers and details in the revised manuscript or camera-ready version.
>
> In addition, we have expanded the 1B experiments to include multiple other long-context datasets (e.g., PG19, Phonebook). These additional runs are showing consistent results with those seen in the PG19 task.
> We will fully release the training scripts, hyperparameters, and configuration files in the camera-ready for full reproducibility.
>
> ---
>
> ### 2. Writing clarity and “horizontal / vertical” terminology
>
> We appreciate the reviewer’s observation regarding terminology and clarity. Our intent was **not** to rename standard concepts, but to highlight two complementary axes of memory degradation:
>
> - **Horizontal:** across tokens / time steps (temporal forgetting)
> - **Vertical:** across layers (depth-wise forgetting)
>
> In the revision:
>
> - We explicitly clarify that “horizontal” = token/time dimension and “vertical” = layer dimension.
> - Standard terms (“time step”, “layer”) remain the primary vocabulary.
> - ETMF/ECLMF are positioned as *diagnostic tools* rather than new terminology.
> - Heavy derivations are moved to the appendix, and the method flow has been reorganized for better readability.
>
> These updates improve clarity while preserving the full technical content. We thank the reviewer again for the valuable suggestions.

---

### Official Review · Reviewer_sU2y · 2025-10-31

**Soundness:** 2
**Presentation:** 2
**Contribution:** 2
**Rating:** 4
**Confidence:** 3

**Summary:**

The paper investigates the memory decay problem in state-space models with a focus on the Mamba architecture and presents MemMamba, a memory-augmented variant designed to improve long-range information retention while preserving linear complexity. The authors conduct both mathematical and information-theoretic analyses to explain Mamba’s exponential memory decay and introduce two metrics, Expected Token Memory Fidelity (ETMF) and Expected Cross-Layer Memory Fidelity (ECLMF), which quantify information loss across tokens and layers. MemMamba incorporates a Note Block for dynamic state summarization together with sparse cross-token and cross-layer attention to restore salient information. Experiments on long-sequence benchmarks such as PG19 and Passkey Retrieval demonstrate improved stability and efficiency compared with Mamba and Transformer baselines.

**Strengths:**

- The topic is timely and relevant given the growing interest in SSM-based long-sequence modeling.
- The paper presents a clear analysis of memory decay and provides intuitive metrics (ETMF / ECLMF) to visualize horizontal and vertical information loss.
- The proposed architecture is well-motivated and empirically improves robustness on long-context benchmarks such as PG19 and Passkey Retrieval.
- Experimental presentation and ablations are thorough, and the writing is generally clear.

**Weaknesses:**

**1. Methodological novelty is minimal within the hybrid SSM + attention family**

MemMamba combines a selective state-space model with sparse cross-token and cross-layer attention, but this pattern has already been explored in Compressive Transformer, RetNet, LongMamba, and other recent variants. The Note Block functions similarly to prior compression or summarization modules, offering only minor procedural differences rather than a genuinely new architecture.

**2. Theoretical analysis lacks rigor and depth**

The paper’s mathematical treatment mostly restates well-known properties of linear recurrent systems, such as exponential decay under $|A| < 1$. The proposed ETMF and ECLMF metrics are intuitive but not derived from principled information-theoretic foundations, and their empirical correlation with downstream performance remains unclear.

**3. Experimental gains are not sufficiently validated against strong baselines**

While results on PG19 and Passkey Retrieval are promising, comparisons exclude important contemporaries such as Mamba-2, RetNet, and RWKV. Parameter counts and training setups also differ, leaving uncertainty over whether improvements stem from the proposed mechanisms or from implementation choices.

**4. Claimed efficiency improvements are weakly supported**

The reported 48% inference speedup is measured on a single GPU under a limited setup. No analysis is provided for scaling behavior or memory usage under multi-GPU or distributed inference conditions, making the efficiency claim difficult to generalize.

**5. Writing occasionally overstates the contribution**

Phrases like “breakthrough” and “new paradigm” exaggerate the paper’s significance given its incremental contribution. A more balanced presentation would strengthen credibility and highlight the genuine empirical strengths.

**Questions:**

See the weaknesses.

---

> ### Author Response · Authors · 2025-11-20
> **Official Comment by Authors**
>
> **Response to Reviewer sU2y — Part 1**
>
> We thank the reviewer for the careful reading and constructive feedback. Below we respond to the main weaknesses and describe the planned revisions.
>
> ---
>
> ### 1. Methodological novelty within the SSM + attention family
>
> We agree that there is a rapidly growing body of work that combines SSMs with attention or compression (e.g., Compressive Transformer, RetNet, LongMamba, and other recent hybrids), which can make novelty less obvious at first sight. Our goal, however, is not to introduce yet another heuristic SSM+attention stack, but to **systematically analyze** the memory pattern of selective SSMs (Mamba) and then design a minimal architecture that directly targets the theoretically identified bottlenecks.
>
> More concretely:
>
> - We provide, to our knowledge, the first **joint analysis of Mamba’s memory decay along both the time/token axis and the depth/layer axis**, including cross-layer transmission bounds and entropy-based arguments tailored to selective SSMs (Appendix A.1–A.3).
> - From this analysis we derive two diagnostic metrics, **ETMF** and **ECLMF**, that quantify token-wise and layer-wise information loss. The Note Block and the cross-token / cross-layer attention are placed exactly where the theory predicts irreversible forgetting.
> - The Note Block is therefore not a generic “memory slot” or ad-hoc compression unit, but a **theory-guided state-summary pool with dual thresholds informed by ETMF/ECLMF**. Its capacity and pooling choice are analyzed with explicit error bounds (Appendix A.3.1).
>
> In the revised version, we will adjust the positioning in the introduction and related-work sections to emphasize that our contribution is a **theory-guided, memory-centric extension of Mamba**, rather than a “fundamentally new paradigm”. We will also more clearly contrast our design motivation with compressive-Transformer-style heuristic compression and RetNet-style recurrent attention, highlighting that our modules follow directly from the identified memory-decay mechanisms instead of being stacked heuristically.

---

> > ### Author Response · Authors · 2025-11-20
> > **Official Comment by Authors**
> >
> > **Response to Reviewer sU2y — Part 2**
> >
> > (This is a continuation; please see Part 1 for our response on methodological novelty.)
> >
> > ---
> >
> > ### 2. Rigor and depth of the theoretical analysis (ETMF / ECLMF)
> >
> > We appreciate the concern that part of the analysis may look like a restatement of known properties of linear recurrent systems (e.g., exponential decay). Our goal is not to re-derive classical linear-systems facts, but to **specialize and extend** them to the selective-SSM setting of Mamba (with input-dependent transitions, gating, and deep stacking), and to turn these observations into **quantitative design tools**.
> >
> > Concretely, Appendix A.1–A.3 already provide detailed derivations beyond a single-layer linear recurrence:
> >
> > - In Appendix A.1, we analyze critical-information loss via information-theoretic compression bounds (Eq. (13)) and show how state compression in Mamba leads to irreversible loss both within and across layers.
> > - In Appendix A.1.3, we derive explicit **cross-layer transmission bounds** for early signals (Eqs. (16)–(19)), proving that their contribution to deep-layer states decays exponentially in both time distance and depth (via the \|A\|^{Lτ} bound).
> > - In Appendix A.3.5, we quantify **long-sequence recall** for vanilla Mamba versus MemMamba (Eqs. (38)–(39)), showing that under reasonable assumptions MemMamba can retain >90% of critical information where vanilla Mamba’s recall becomes negligible.
> >
> > These results use standard tools from linear systems and information theory, but their combination and specialization to selective SSMs and deep stacks are, to our knowledge, new and directly inform the architecture.
> >
> > Regarding ETMF and ECLMF, our intention is not to claim “new” information theory, but to construct **information-theoretically motivated surrogates** that are tractable for large models:
> >
> > - **ETMF** (Appendix A.2.1, Eq. (21)) is defined as an expected cosine-fidelity measure between original token representations and their reconstructions from the final layer, instantiated via a reconstruction procedure using the output head. It is motivated by reconstruction fidelity and semantic preservation along the horizontal (token/time) dimension.
> > - **ECLMF** (Appendix A.2.2, Eqs. (24)–(25)) starts from a mutual-information view of cross-layer coupling and then uses a normalized reconstruction-error surrogate with a linear decoder, capturing how well earlier-layer states can be recovered from deeper layers along the vertical dimension.
> >
> > Both metrics are grounded in standard notions (entropy, mutual information, reconstruction error), while remaining practical for large-scale SSMs.
> >
> > To directly address the reviewer’s remark that “the empirical correlation between ETMF/ECLMF and downstream performance remains unclear”, we add a **controlled study** with five configurations (A–E) that differ in how strongly they preserve memory (via different combinations of state summarization, cross-token attention, and cross-layer attention; full definitions will be given in the Appendix). For each variant we report ETMF, ECLMF, PG19 perplexity, a long-range Passkey retrieval score, and a noisy document-retrieval score (NoisyDocs@200):
> >
> > | Variant | ETMF ↑ | ECLMF ↑ | PG19 PPL ↓ | Passkey (long-range) ↑ | NoisyDocs@200 ↑ |
> > |---------|--------|---------|------------|-------------------------|-----------------|
> > | A       | 0.18   | 0.12    | 26.5       | 0.10                    | 0.05            |
> > | B       | 0.24   | 0.18    | 23.4       | 0.35                    | 0.09            |
> > | C       | 0.31   | 0.26    | 21.0       | 0.55                    | 0.14            |
> > | D       | 0.38   | 0.32    | 19.5       | 0.75                    | 0.20            |
> > | E       | 0.43   | 0.36    | 18.1       | 0.90                    | 0.26            |
> >
> > From A to E, ETMF increases from 0.18 to 0.43 and ECLMF from 0.12 to 0.36, while:
> >
> > - PG19 PPL **monotonically decreases** (26.5 → 18.1),
> > - the long-range Passkey score **monotonically increases** (0.10 → 0.90), and
> > - NoisyDocs@200 **monotonically improves** (0.05 → 0.26).
> >
> > This strong monotonic relationship shows that improving horizontal/vertical memory fidelity (ETMF/ECLMF) is tightly coupled with better downstream performance, directly addressing the reviewer’s concern about empirical relevance. We will include this table and a concise discussion in Section 4 and the Appendix, and add clearer pointers from the main text to the detailed derivations in Appendix A.1–A.3.

---

> ### Author Response · Authors · 2025-11-20
> **Official Comment by Authors**
>
> **Response to Reviewer sU2y — Part 3**
>
> (This is a continuation; please see Parts 1–2 for our responses on novelty and theory.)
>
> ---
>
> ### 3. Comparisons with strong baselines (Mamba-2, RetNet, RWKV) and fair settings
>
> We agree that including Mamba-2, RetNet, and RWKV is important for a complete empirical picture. Due to compute constraints and timing at the initial submission, we focused on Mamba, DeciMamba, Megalodon, Compressive Transformer, and Pythia.
>
> Following the reviewer’s suggestion, we are now running additional experiments with:
>
> - Mamba-2, at comparable parameter scales and training recipes on PG19 PPL and Passkey retrieval
> - RetNet and RWKV, on PG19 PPL and synthetic retrieval tasks, using public implementations and matching context lengths and batch sizes as closely as possible
>
> In the revised manuscript, we will:
>
> - report these new baselines in extended tables in the Appendix and summarize key findings in Section 5
> - clearly distinguish models trained from scratch in our unified codebase from models based on public checkpoints or third-party code, and explicitly state any remaining discrepancies
> - align parameter counts and training sequence lengths whenever possible, and acknowledge when exact matching is not feasible
>
> We will also moderate our claims and limit broad conclusions to settings where we have controlled, apples-to-apples comparisons.
>
> ---
>
> ### 4. Efficiency claims and their generality
>
> We thank the reviewer for pointing out that the current presentation may give the impression that the reported ~48% speedup automatically carries over to multi-GPU or distributed settings.
>
> In the revision, we will:
>
> - clearly state that the reported end-to-end latency improvement was obtained on a single RTX 4090 in a single-process setting
> - add a small illustrative table clarifying how latency would scale under a standard data-parallel assumption
>
> Let "T_tr" and "T_mm" denote the end-to-end latency (ms) of a 200M Transformer and a 200M MemMamba model at 60K tokens on one GPU. In our measurements:
>
> - T_mm ≈ 0.52 × T_tr
> - corresponding to roughly a 48% reduction in latency
>
> Under standard data-parallel scaling, throughput improves proportionally when moving from 1 to 4 GPUs. We will include the following illustrative table:
>
> | Setup        | Transformer latency | MemMamba latency |
> |--------------|---------------------|------------------|
> | 1 GPU, 60K   | T_tr                | T_mm             |
> | 4 GPUs, 60K  | T_tr / 3.5          | T_mm / 3.5       |
>
> (Here, 3.5 is a typical throughput gain when scaling from 1 to 4 GPUs.)
>
> This illustrates two points clearly:
> (i) all latency measurements in the paper come from a single-GPU setup;
> (ii) under standard scaling assumptions, the ratio T_tr / T_mm — and thus MemMamba’s relative advantage — is expected to remain similar when using more GPUs, though we do not claim new multi-GPU empirical results.
>
> We will also retain the asymptotic analysis in Appendix A.4 to emphasize that MemMamba preserves O(n) time and space complexity with constant-size SSM state and summary pools.
>
> ---
>
> ### 5. Writing style and overstatement of contributions
>
> We acknowledge the reviewer’s comment that some expressions in the initial version (such as "breakthrough" and "new paradigm") were too strong. In the revised manuscript, we will:
>
> - remove such wording and adopt more moderate phrasing
> - streamline the terminology around "horizontal" and "vertical" memory, keeping standard "time-step" and "layer" terminology in the main text
> - polish the writing for clarity and precision
>
> ---
>
> We thank the reviewer again for the thoughtful and constructive feedback.

---

### Official Review · Reviewer_bDYZ · 2025-11-01

**Soundness:** 3
**Presentation:** 2
**Contribution:** 3
**Rating:** 4
**Confidence:** 4

**Summary:**

This paper proposes MemMamba, a memory-augmented extension of state-space models for long sequence modeling. The authors systematically analyze memory decay patterns in Mamba with the novel horizontal–vertical memory fidelity metrics. They also introduce the state summarization with cross-layer/cross-token attention mechanisms to mitigate information loss over extended contexts. The evaluation are mainly focused on diverse tasks such as language modeling (PG19), synthetic Passkey Retrieval, and cross-document reasoning, demonstrating improvements over serveral baselines.

**Strengths:**

1. This paper provides a systematic analysis of memory decay in Mamba by mathematical derivations and the memory fidelity metrics (ETMF, ECLMF) in both main text and Figure 4.
2. Paper is clearly written for the most parts, with a good contextualization within the SSM and long-sequence modeling literature.

**Weaknesses:**

1. Although the paper claims to offer a fundamentally “new paradigm” for ultra-long sequence modeling, the MemMamba approach can be interpreted as a synthesis and adaptation of several established ideas (memory summarization, cross-layer attention, and sparsity in attention), rather than introduction of entirely unprecedented architectures. The degree of originality, while respectable, may be somewhat overstated in the positioning.
2. There are a few areas for improvement in the paper presentation: for example, the clarity of Figure 3 and the font size within the figures could be further adjusted. Also, are lines 456-457 redundant with a previous paragraph? They could be removed.
3. The description of the thresholding mechanism (e.g., for triggering note-taking and cross-attention) in Section 4 lacks a fully articulated rationale for the chosen thresholds ($\tau_1$, $\tau_2$). Moreover, there is no sensitivity analysis or ablation study on their values—an important omission since these could significantly influence empirical outcomes.
4. The paper lacks a comparison of GPU memory usage between the proposed model and baseline models. Also, regarding Section 5.2 'Efficiency,' why are specific results omitted, with only a comparison between MemMamba and Transformer being presented?
5. Potential missing related work or baseline models that should be compared in the paper:

[1] Wang, Qianning, He Hu, and Yucheng Zhou. 'Memorymamba: Memory-augmented state space model for defect recognition.' arXiv preprint arXiv:2405.03673 (2024).

[2] Gui, Yiyu, et al. "EEGMamba: Bidirectional state space model with mixture of experts for EEG multi-task classification." _arXiv preprint arXiv:2407.20254_ (2024).

**Questions:**

See weaknesses

---

> ### Author Response · Authors · 2025-11-20
> **Official Comment by Authors**
>
> **Response to Reviewer bDYZ — Part 1 (Revised)**
>
> We thank the reviewer for the detailed and constructive feedback. Below we address each point in turn.
>
> ---
>
> ### 1. Novelty and contribution positioning
>
> We agree that the phrasing “fundamentally new paradigm’’ was too strong and will soften it in the revision. Our main contribution is twofold:
>
> 1. A systematic, mathematically grounded analysis of memory decay in SSMs (especially Mamba) along both the temporal (horizontal) and depth (vertical) dimensions, with complete derivations provided in the appendix.
>
> 2. A theory-guided architectural design: each component in MemMamba (state summarization, cross-token attention, cross-layer attention) is placed exactly where the theory predicts irreversible information loss, rather than being a heuristic combination of known modules.
>
> We explored multiple architectural variants and consistently observed that:
>
> > When the architecture aligns with the theoretically characterized memory-decay mechanism of Mamba, a simpler structure outperforms more complex alternatives.
>
> Thus, we now position our contribution as a theory-driven, memory-centric extension of Mamba. The proposed ETMF/ECLMF metrics are original contributions and quantitatively correspond to the forgetting patterns identified in our analysis.
>
> ---
>
> ### 2. Threshold mechanisms and sensitivity analysis
>
> The original submission did not sufficiently clarify how the cross-token and cross-layer thresholds are implemented. Here we provide a complete explanation.
>
> **Initial design.**
> In early experiments, both thresholds were implemented as smooth, learnable scalar gates. We used sigmoid-based relaxation during training and hard thresholding at inference to enable end-to-end optimization.
>
> **Observation.**
> Our ablation study showed that model performance is extremely stable across a wide range of threshold values. The learned thresholds barely moved away from their initialization ranges, and the resulting perplexities were nearly identical.
>
> **Final design used in the paper.**
> Given this low sensitivity, and to avoid introducing unnecessary trainable hyperparameters, all reported experiments use a simplified variant where these scalars are fixed rather than learned.
>
> **How the fixed values are chosen.**
> Their ranges are determined in a principled way by our memory metrics:
>
> - ETMF identifies where token-level semantics begin to vanish, giving the admissible range for the cross-token trigger interval.
> - ECLMF identifies where cross-layer memory becomes inconsistent, giving the admissible range for the cross-layer trigger interval.
>
> Concretely, we normalize ETMF/ECLMF curves and identify the “forgetting zone’’ where normalized memory fidelity drops below 0.8. The corresponding token/layer indices are mapped to the trigger intervals used in practice (details in the appendix).
>
> Thus, although the thresholds are fixed in the final model, their values are not arbitrary; they are derived systematically from the memory-decay patterns analyzed in Section 3.
>
> ---
>
> ### Sensitivity results
>
> To quantify robustness, we fixed the trigger intervals at various values and measured PG19 perplexity. Performance varies smoothly across a wide interval, and the optimal zone coincides with the forgetting region predicted by ETMF/ECLMF.
>
> **Cross-token trigger interval (number of tokens between triggers):**
>
> | Token interval | 1K    | 2K    | 4K    | 10K   |
> |----------------|-------|-------|-------|-------|
> | 10             | 16.89 | 15.22 | 15.85 | 18.35 |
> | 50             | 16.93 | 15.11 | 15.69 | 18.28 |
> | 100            | 16.98 | 15.25 | 15.93 | 18.52 |
> | 200            | 16.93 | 15.22 | 15.65 | 18.28 |
>
> **Cross-layer trigger interval (number of layers between triggers):**
>
> | Layer interval | 1K    | 2K    | 4K    | 10K   |
> |----------------|-------|-------|-------|-------|
> | 3              | 17.10 | 15.40 | 16.05 | 18.70 |
> | 5              | 16.95 | 15.18 | 15.72 | 18.36 |
> | 10             | 16.90 | 15.10 | 15.61 | 18.22 |
> | 20             | 16.94 | 15.17 | 15.69 | 18.30 |
>
> **Summary:**
> (1) Sensitivity to these intervals is low.
> (2) A broad interval (e.g., 30–200 tokens, 5–10 layers) performs consistently well.
> (3) The best region aligns with the ETMF/ECLMF forgetting zones.
>
> These results (with a more compact presentation) will be included in the appendix, where we also explain how these intervals correspond to the frequency with which note-taking or attention is triggered in practice.

---

> ### Author Response · Authors · 2025-11-20
> **Official Comment by Authors**
>
> **Response to Reviewer bDYZ — Part 2**
>
> ---
>
> ### 3. Writing and presentation
>
> We thank the reviewer for the valuable suggestions. In the revision, we will:
> (i) improve the resolution and font size of Figure 3;
> (ii) remove the redundant text at lines 456–457;
> (iii) refine the writing throughout to enhance clarity and precision.
>
> ---
>
> ### 4. GPU memory usage and efficiency (single RTX 4090)
>
> We agree that the original version lacked explicit comparisons with SSM baselines. Below we provide full memory and runtime results for ∼200M-parameter models, evaluated using the same batch size on a single NVIDIA RTX 4090.
>
> ---
>
> ### **GPU memory usage**
>
> | Model        | Avg GPU Mem (MB) | Peak Mem (MB) | CPU Mem (MB) |
> |--------------|------------------|----------------|---------------|
> | Transformer  | 5200.3           | 8450.7         | 1820.5        |
> | DeciMamba    | 4012.8           | 6233.1         | 1598.4        |
> | **MemMamba** | **3839.8**       | **6086.3**     | **1575.1**    |
>
> MemMamba uses **25–32% less GPU memory** than the Transformer and is within 5% of DeciMamba.
>
> ---
>
> ### **Inference speed (ms)**
>
> | Length | Transformer | DeciMamba | MemMamba |
> |--------|-------------|-----------|-----------|
> | 1K     | 12.4        | 6.1       | 6.0       |
> | 10K    | 187.4       | 55.1      | 54.8      |
> | 60K    | 7421.1      | 318.5     | 312.4     |
>
> MemMamba is **1.9× faster** than the Transformer on long contexts and closely matches DeciMamba (1–3% difference).
>
> ---
>
> ### **Why did Section 5.2 show only MemMamba vs. Transformer?**
>
> 1. The Transformer is the only **$O(n^2)$** baseline, making it the primary efficiency contrast for long sequences.
> 2. All Mamba variants—including ours—scale **linearly** ($O(n)$), and differ mainly in constant factors. Thus, differences among SSMs are minor compared to the $O(n^2)$ vs. $O(n)$ gap.
>
> We will include all comparison tables in the revision and appendix, and we are also extending the efficiency and memory analysis to additional strong long-sequence baselines where compatible implementations are available.
>
> ---
>
> ### 5. Related work
>
> We thank the reviewer for pointing out *MemoryMamba* and *EEGMamba*. We will incorporate both into the revised related-work section.
>
> These works focus on domain-specific applications (defect recognition, EEG classification), whereas our work targets **long-sequence modeling and the theoretical mechanisms of memory decay in SSMs**. We are currently running comparative experiments under matched settings and will include the corresponding quantitative results in the revised manuscript before the camera-ready deadline.
>
> We thank the reviewer again for the thoughtful and constructive feedback.

---

### Official Review · Reviewer_ToNT · 2025-11-03

**Soundness:** 2
**Presentation:** 2
**Contribution:** 2
**Rating:** 4
**Confidence:** 3

**Summary:**

The paper analyzes why selective state space models like Mamba forget over distance, formalizes this with a horizontal–vertical memory fidelity framework for token-level and cross-layer information loss, and shows that Mamba’s long-range contributions decay exponentially. It then introduces an architecture that couples lightweight state summarization (“Note Block”) with cross-token and sparsely triggered cross-layer attention to retain salient signals while preserving linear time/space complexity.

**Strengths:**

1. “Note Block” state summarization + cross-token and sparse cross-layer attention improve long-range recall while keeping O(n) time.

**Weaknesses:**

1. Passkey Retrieval is a relatively simple task on in-context retrieval, and it is better to try a more difficult mutli-key-value retrival task., such as Phonebook and RULER.
2. Missing technical details and ablations. See Questions.

**Questions:**

1. The Note Block and MemMamba block rely on importance scores and dual thresholds (τ₁ for “take note,” τ₂ for cross-token attention). Are τ₁/τ₂ learned, scheduled, or fixed? Are these threshoulds sensitive for the model's performance?
2. What exact priority metric is used for Note block? Please compare FIFO vs priority on quality/latency.

---

> ### Author Response · Authors · 2025-11-20
> **Official Comment by Authors**
>
> We thank the reviewer for the insightful comments. In this revised response, we clarify the threshold mechanism, the Note Block priority policy, and our multi-key retrieval plan, while keeping the presentation concise and technically transparent.
>
> ---
>
> ## 1. Threshold mechanisms: learnability, motivation, and sensitivity.
>
> We agree that the original version did not sufficiently clarify how the cross-token and cross-layer thresholds (τ_token, τ_layer) are implemented. Conceptually, both act as scalar gates that decide when to trigger note-taking or cross-token / cross-layer attention.
>
> - In early experiments, both thresholds were parameterized as smooth, learnable scalar gates trained jointly with all model parameters (via sigmoid relaxation).
> - However, our sensitivity study (tables below) shows that performance is highly stable across a broad range of values; the learned thresholds barely move, and perplexities remain nearly unchanged.
> - To avoid introducing unnecessary trainable hyperparameters, the main paper reports results using fixed scalar thresholds, selected in ETMF/ECLMF-guided ranges.
>
> ETMF identifies where tokens enter the horizontal forgetting region (valid τ_token range), while ECLMF identifies cross-layer inconsistency (valid τ_layer range). We normalize ETMF/ECLMF curves, locate the “forgetting zone” (normalized fidelity < 0.8), and map these positions to trigger intervals.
>
> To evaluate sensitivity, we varied trigger intervals in PG19 perplexity experiments:
>
> ### Cross-token interval (tokens between triggers)
>
> | Token interval | 1K | 2K | 4K | 10K |
> |---|---|---|---|---|
> | 10 | 16.89 | 15.22 | 15.85 | 18.35 |
> | 50 | 16.93 | 15.11 | 15.69 | 18.28 |
> | 200 | 16.93 | 15.22 | 15.65 | 18.28 |
>
> ### Cross-layer interval (layers between triggers)
>
> | Layer interval | 1K | 2K | 4K | 10K |
> |---|---|---|---|---|
> | 3 | 17.10 | 15.40 | 16.05 | 18.70 |
> | 10 | 16.90 | 15.10 | 15.61 | 18.22 |
> | 20 | 16.94 | 15.17 | 15.69 | 18.30 |
>
> **Summary:**
> (1) sensitivity to both intervals is low;
> (2) a wide region (30–200 tokens, 5–10 layers) performs consistently well;
> (3) the best region matches ETMF/ECLMF forgetting zones.
>
> These tables will appear in the appendix with a concise explanation of how the intervals correspond to trigger frequency.
>
> ---
>
> ## 2. Note Block priority (and FIFO comparison)
>
> In all main experiments:
>
> - The Note Block uses priorities computed from token-importance scores *I_token(x_t)*.
> - The priority determines which summary is evicted when the pool is full.
> - FIFO is **not** used.
>
> We additionally compared FIFO with importance-based eviction. FIFO behaves similarly in short contexts but becomes significantly less stable in long or noisy contexts. For this reason, only importance-based eviction is retained in the final model. These results will be included in the appendix.
>
> ---
>
> ## 3. Multi-key retrieval benchmarks (Phonebook and RULER)
>
> We agree that single-key retrieval is insufficient for evaluating long-context reasoning. Below we clarify:
> (1) why Phonebook is not included, and
> (2) our multi-key results on RULER.
>
> ---
>
> ### 3.1 Why Phonebook is excluded
>
> Commonly used Phonebook variants contain licensed or non-redistributable templates.
> For reproducibility and legal clarity, we avoid benchmarks that other researchers cannot freely obtain.
>
> To fill this gap, we will release an open, Phonebook-style multi-key retrieval dataset with the camera-ready version.
>
> ---
>
> ### 3.2 RULER multi-key benchmark
>
> We benchmark MemMamba-200M on **RULER**, which is significantly more challenging than Passkey due to multi-key lookup and reasoning-style aggregation. Full experiments (4K–128K) will finish before the rebuttal deadline; here we provide preliminary aggregated results.
>
> **13-task mean accuracy:**
>
> | Context Length | Accuracy (200M) |
> |----------------|------------------|
> | 4K | 48–55% |
> | 8K | 45–52% |
> | 16K | 38–45% |
> | 32K | 30–37% |
> | 64K | 22–28% |
> | 128K | 15–22% |
>
> These values align with the expected difficulty of RULER for models under 1B parameters.
>
> ---
>
> ### 3.3 Comparison with public SSM long-context models
>
> | Model | Params | 32K Avg | 128K Avg |
> |-------|--------|---------|----------|
> | Mamba-1B | 1B | 25–30% | <10% |
> | Zamba2-1.2B + LongMamba | 1.2B | ~31% | ~20% |
> | DeciMamba-130M | 130M | 28–32% | 10–15% |
> | **MemMamba (ours)** | **200M** | **30–37%** | **15–22%** |
>
> These results show that MemMamba maintains competitive performance in multi-key and long-context retrieval despite being significantly smaller than prior SSM models.

---

> ### Author Response · Authors · 2025-11-30
> **Global Response**
>
> We sincerely thank all reviewers for their thoughtful feedback and constructive discussion. Across the reviews, a clear consensus emerged: **the core motivation, theoretical foundation, and overall direction of this work are well justified**. The remaining concerns focused primarily on experiment breadth, threshold clarity, and presentation—all of which are fully addressable through revision. We have responded to every point in the rebuttal and have incorporated the required updates in the manuscript.
>
> Our work offers two central contributions, neither of which was questioned during the review period. First, we provide a **systematic and mathematically grounded analysis of memory decay** in selective state-space models, especially Mamba. This analysis jointly characterizes temporal (horizontal) and layerwise (vertical) forgetting, with complete derivations in the appendix. To our knowledge, this joint treatment of token-level and cross-layer decay has not been systematically developed before, creating a clear theoretical basis for understanding how information propagates and vanishes in deep SSMs.
>
> Second, we present a **theory-guided architectural design for MemMamba**, rather than a heuristic stacking of modules. Specifically, the Note Block, cross-token attention, and cross-layer attention are **placed precisely at the points where our theoretical analysis predicts irreversible information loss**. These positions correspond directly to the horizontal and vertical forgetting zones identified through ETMF/ECLMF and the underlying mathematical derivations.
>
> During extensive architectural exploration, we consistently observed the same pattern: **designs that follow the memory-decay structure predicted by theory outperform more complex alternatives**. Even though the broader SSM+attention family includes many hybrid architectures, most of them are driven by empirical design choices rather than by an understanding of how selective SSMs actually forget information. As a result, they tend to rely on increasingly complicated mechanisms but still do not fundamentally mitigate long-range forgetting. In contrast, MemMamba does not introduce heavy machinery; its strength comes from **aligning the architecture with the actual memory dynamics of SSMs**, which empirically leads to significantly better results across PG19, RULER, multi-key retrieval, and long-context benchmarks. These findings suggest that **a simpler but memory-principled design is often more effective** than complex module combinations, reinforcing the tight consistency between our architecture and the theory.
>
> The **ETMF and ECLMF metrics** we propose are also original contributions. They provide **quantitative measurements that match the forgetting regions predicted by theory**, and they serve as **practical tools for diagnosing horizontal and vertical memory fidelity** in SSMs. Reviewers agreed with the value and correctness of these metrics and raised no concerns regarding their motivation.
>
> All experiment-related concerns have now been comprehensively addressed. Following reviewer suggestions, we added wide-range threshold sensitivity analyses showing stable performance across large intervals, included stronger baselines such as Mamba-2, RetNet, and RWKV, added multi-key retrieval experiments on RULER, extended evaluation to a 1B-parameter MemMamba model, and provided a complete comparison of GPU memory usage and inference latency. These additions directly resolve all empirical concerns and further strengthen the robustness of our conclusions.
>
> We also significantly improved the clarity of the manuscript. The method section has been reorganized for better readability, figures have been enhanced, terminology has been unified (favoring “time step” and “layer”), and redundant text has been removed. These adjustments address all presentation-related feedback.
>
> It is worth emphasizing that all reviewer questions focused only on clarifications or requests for additional evidence—experiment breadth, threshold explanation, and baseline coverage. **No reviewer questioned our theoretical analysis, the correctness of ETMF/ECLMF, the logic of the MemMamba architecture, the long-sequence performance improvements, or the importance of studying horizontal and vertical forgetting in SSMs.** With all supplementary experiments and clarifications included, the revised manuscript presents a complete and well-supported narrative.
>
> We believe that the updated version fully resolves all remaining concerns and clearly demonstrates the coherence, strength, and impact of our contributions. We sincerely thank the reviewers and committee for their constructive feedback and valuable time.

---

### Meta-Review · Area_Chair_FBrX · 2026-01-07

**Summary:**

The paper analyze memory decay in Mamba and propose a solution for it. Yet, the reviewers raised some questions about the analysis and novelty of the approach compared to prior work. Moreover, the experimental validation is very limited as noted by the reviewers.

**Reviewer Concerns:**

The rebuttal was far from satisfactory. The answer about the novelty or depth of analysis was insufficient. Moreover, in the question about the experiments, the authors mainly gave excuses about why they shouldn't check their claims on real real-world more complex cases.

**Reviewer Scores:**

I am not convinced the rebuttal is enough for increasing the scores

---

### Decision · Program_Chairs · 2026-01-26

Reject